# PHF3 regulates neuronal gene expression through the Pol II CTD reader domain SPOC

Lisa-Marie Appel [1,13], Vedran Franke [2,13], Melania Bruno [1,13], Irina Grishkovskaya[3,13], Aiste Kasiliauskaite[4,5], Tanja Kaufmann [1], Ursula E. Schoeberl[5], Martin G. Puchinger[3], Sebastian Kostrhon[1], Carmen Ebenwaldner[1], Marek Sebesta[4], Etienne Beltzung[1], Karl Mechtler [6,7], Gen Lin[6], Anna Vlasova[6], Martin Leeb [8], Rushad Pavri[6], Alexander Stark [6], Altuna Akalin [2], Richard Stefl[4,5], Carrie Bernecky [9], Kristina Djinovic-Carugo [3,10] & Dea Slade [1,11,12✉]

The C-terminal domain (CTD) of the largest subunit of RNA polymerase II (Pol II) is a regulatory hub for transcription and RNA processing. Here, we identify PHD-finger protein 3 (PHF3) as a regulator of transcription and mRNA stability that docks onto Pol II CTD through its SPOC domain. We characterize SPOC as a CTD reader domain that preferentially binds two phosphorylated Serine-2 marks in adjacent CTD repeats. PHF3 drives liquid-liquid phase separation of phosphorylated Pol II, colocalizes with Pol II clusters and tracks with Pol II across the length of genes. PHF3 knock-out or SPOC deletion in human cells results in increased Pol II stalling, reduced elongation rate and an increase in mRNA stability, with marked derepression of neuronal genes. Key neuronal genes are aberrantly expressed in Phf3 knock-out mouse embryonic stem cells, resulting in impaired neuronal differentiation. Our data suggest that PHF3 acts as a prominent effector of neuronal gene regulation by bridging transcription with mRNA decay.

[1] Department of Biochemistry and Cell Biology, Max Perutz Labs, University of Vienna, Vienna Biocenter (VBC), Vienna, Austria. [2] The Berlin Institute for Medical Systems Biology, Max Delbrück Center, Berlin, Germany. [3] Department of Structural and Computational Biology, Max Perutz Labs, University of Vienna, Vienna Biocenter (VBC), Vienna, Austria. [4] CEITEC-Central European Institute of Technology, Masaryk University, Brno, Czech Republic. [5] National Centre for Biomolecular Research, Faculty of Science, Masaryk University, Brno, Czech Republic. [6] Research Institute of Molecular Pathology (IMP), Campus-Vienna-Biocenter 1, Vienna Biocenter (VBC), Vienna, Austria. [7] Institute of Molecular Biotechnology of the Austrian Academy of Sciences (IMBA), Vienna Biocenter (VBC), Vienna, Austria. [8] Department of Microbiology, Immunobiology and Genetics, Max Perutz Labs, University of Vienna, Vienna Biocenter (VBC), Vienna, Austria. [9] Institute of Science and Technology Austria (IST Austria), Am Campus 1, Klosterneuburg, Austria. [10] Department of Biochemistry, Faculty of Chemistry and Chemical Technology, University of Ljubljana, Ljubljana, Slovenia. [11] Department of Radiation Oncology, Medical University of Vienna, Vienna, Austria. [12] Comprehensive Cancer Center, Medical University of Vienna, Vienna, Austria. [13] These authors contributed equally: Lisa-Marie Appel, Vedran Franke, Melania Bruno, Irina Grishkovskaya. ✉email: dea.slade@maxperutzlabs.ac.at

Transcription is a highly regulated process of RNA Polymerase II (Pol II) recruitment, initiation, pausing, elongation and termination[1–3]. Transcription regulation is critical to establish and maintain cell identity, and transcription misregulation underlies cancer and other diseases[4]. Transcription elongation factors modulate Pol II pause release, backtracking, elongation rate or processivity, and couple transcription elongation with co-transcriptional RNA processing[5–17].

Transcription regulators interact with structurally defined surfaces of the Pol II complex and with the unstructured C-terminal domain (CTD) of the largest Pol II subunit RPB1[18–20]. Heptarepeats ($Y_1S_2P_3T_4S_5P_6S_7$) within the Pol II CTD undergo dynamic phosphorylation during transcription, orchestrating the timely recruitment of regulatory factors. The early stages of transcription are marked by Pol II phosphorylated on Serine-5 (pS5), whereas productive elongation is characterized by the removal of pS5 and a concomitant increase in phosphorylated Serine-2 (pS2)[18]. Transcription is tightly coupled with co-transcriptional RNA processing whereby Pol II CTD acts as a docking site for 5' mRNA capping, splicing, 3'end processing, termination and mRNA export factors that recognize specific CTD phosphorylation patterns[10,21–23]. Yeast and mammalian 5' mRNA capping enzymes such as Cgt1, Pce1 and Mce1 were shown to employ the nucleotidyltransferase (NT) domain to directly bind pS5 within the Pol II CTD, whereas 3'end processing and termination factors such as yeast Pcf11 and mammalian SCAF8 employ the CTD-interaction domain (CID) to directly bind the pS2 mark on the Pol II CTD[24–28]. Pol II CTD undergoes liquid-liquid phase separation (LLPS) as a means of compartmentalizing transcription initiation machinery in unphosphorylated Pol II CTD clusters, whereas phosphorylated CTD partitions with RNA processing factors[29,30].

The PHD-finger protein 3 (PHF3) belongs to a family of putative transcriptional regulators that includes the human Death-Inducer Obliterator (DIDO) and yeast Bypass of Ess-1 (Bye1)[31,32]. This family of proteins contains two motifs found in several transcription factors: a domain that is distantly related to the Pol II–associated domain of the elongation factor TFIIS, called the TFIIS-Like Domain (TLD), and a Plant Homeo Domain (PHD)[31]. It also contains a Spen Paralogue and Orthologue C-terminal (SPOC) domain, which has been associated with cancer, apoptosis and transcription[33]. Similar to TFIIS, Bye1 binds the jaw domain of RPB1 via its TLD in vitro and in vivo[31,34]. In contrast to TFIIS, Bye1 TLD does not stimulate mRNA cleavage during transcriptional proofreading due to the absence of the TFIIS domain III[31,35]. Combined deletion of the PHD and SPOC domains abrogated Pol II binding by Bye1 in vivo, suggesting that the Bye1 TLD is necessary but not sufficient to interact with Pol II[34]. Although PHF3 does not contain any canonical CTD-binding domains, it was recently identified in a mass spectrometry screen for proteins binding to phosphorylated GST-CTD[36]. However, the physiological relevance of this interaction, and whether or how PHF3 regulates transcription, remain unknown.

Here, we discover an unexpected interaction between PHF3 and the Pol II CTD via the SPOC domain. We show that PHF3 SPOC displays specificity towards CTD repeats phosphorylated on S2, establishing the SPOC domain as a reader of the Pol II CTD. Moreover, we find that PHF3 colocalizes with Pol II clusters inside cells and forms condensates in vitro, which incorporate phosphorylated CTD and Pol II. PHF3 exerts a dual regulatory function on both Pol II transcription and mRNA stability in a global and gene-specific manner. Neuronal genes are aberrantly derepressed in PHF3 KO HEK293T cells, and Phf3 KO mESCs show impaired neuronal differentiation. Overall, our data suggest that PHF3, as a regulator of Pol II transcription and mRNA stability via the CTD, controls a neuronal gene expression program.

## Results

**PHF3 interacts with RNA polymerase II through the SPOC domain.** To explore the function of PHF3 in transcription, we expressed FLAG-PHF3 in HEK293T cells and identified interacting proteins by co-immunoprecipitation (co-IP) followed by mass spectrometry (Fig. 1a, b and Supplementary Data 1). RPB1 ranked highest among 40 high-confidence PHF3 interactors (Supplementary Data 1), including other regulators of Pol II transcription elongation (SPT5, SPT6, PAF1C, FACT), as well as RNA processing factors (Fig. 1a, b and Supplementary Data 1). We confirmed these findings by co-IP analyses of endogenous PHF3 tagged with GFP in HEK293T cells, due to the lack of suitable commercially available PHF3 antibodies (Fig. 1c, d and Supplementary Fig. 1a). Further, we found that endogenously expressed PHF3-GFP interacted with Pol II phosphorylated on Serines 2, 5 and/or 7 within the heptarepeats (Fig. 1d, e).

To determine which domains within PHF3 are required for these interactions, we performed a co-IP analysis of different FLAG-PHF3 deletion mutants expressed in HEK293T cells (Fig. 1f). The truncation mutants localized to the nucleus and bound chromatin, similar to full-length PHF3 (Supplementary Fig. 1c, d). However, neither the TLD nor PHD domains were required for the interaction between Pol II and FLAG-PHF3 (Fig. 1f), which was unexpected given that the TLD is required for the Bye1-Pol II interaction in yeast[31,34]. In contrast, removal of the SPOC domain from PHF3 completely abrogated the interaction with Pol II, the elongation factor SPT5 and the Pol II-associated factor PAF1 (Fig. 1f). Moreover, the isolated PHF3 SPOC domain (aa 1199–1356) was sufficient to bind phosphorylated Pol II (Fig. 1g).

**PHF3 SPOC preferentially binds RNA Pol II CTD phosphorylated on Serine-2.** SPOC domains display low sequence conservation, but several amino acids within a positively charged patch on the SPOC surface are highly conserved, including an Arg residue and a Tyr residue (R1248 and Y1291 for PHF3 SPOC, Fig. 2a). These residues are required for the SPOC-containing protein SHARP (SPEN) to interact with the co-repressor complex SMRT/NCoR[37]. Serine phosphorylation of the LSD motif within SMRT or NCoR increases their binding affinity for the conserved Arg within the SHARP SPOC domain, suggesting that the SPOC domain is a phospho-serine binding module[38,39].

Based on these observations, we hypothesized that the positively charged surface of PHF3 SPOC binds the phosphorylated heptarepeats of Pol II CTD. To test this hypothesis, we examined the binding of bacterially expressed PHF3 SPOC to various phosphoisoforms of a CTD diheptapeptide (YSPTSPS-YSPTSPS) in vitro (Supplementary Table 1). PHF3 SPOC did not bind the unphosphorylated CTD diheptapeptide or CTD phosphorylated on only one repeat (Supplementary Fig. 2a), but phosphorylation of S2 within both repeats (2xpS2) was sufficient to confer strong binding (Fig. 2b, c and Supplementary Fig. 2a–f). Comparable affinity of PHF3 SPOC towards 2xpS2 ($K_d = 1.6 \pm 0.3\ \mu M$), 2xpS2pS7 ($K_d = 0.8 \pm 0.1\ \mu M$) and 2xpS2pS5 ($K_d = 4.8 \pm 0.3\ \mu M$), coupled with lower affinity for 2xpS5 ($K_d = 20.0 \pm 4.0\ \mu M$) or 2xpS7 ($K_d = 26.0 \pm 2.9\ \mu M$), suggested that PHF3 SPOC preferentially binds 2xpS2 (Fig. 2b, c and Supplementary Fig. 2a–f). The requirement of tandem pS2 phosphorylation marks for stable binding of PHF3 SPOC to Pol II CTD is in line with genetic studies in yeast showing that the minimal functional unit of CTD is a diheptad[40,41], and with mass spectrometry analysis of CTD phosphorylation patterns revealing

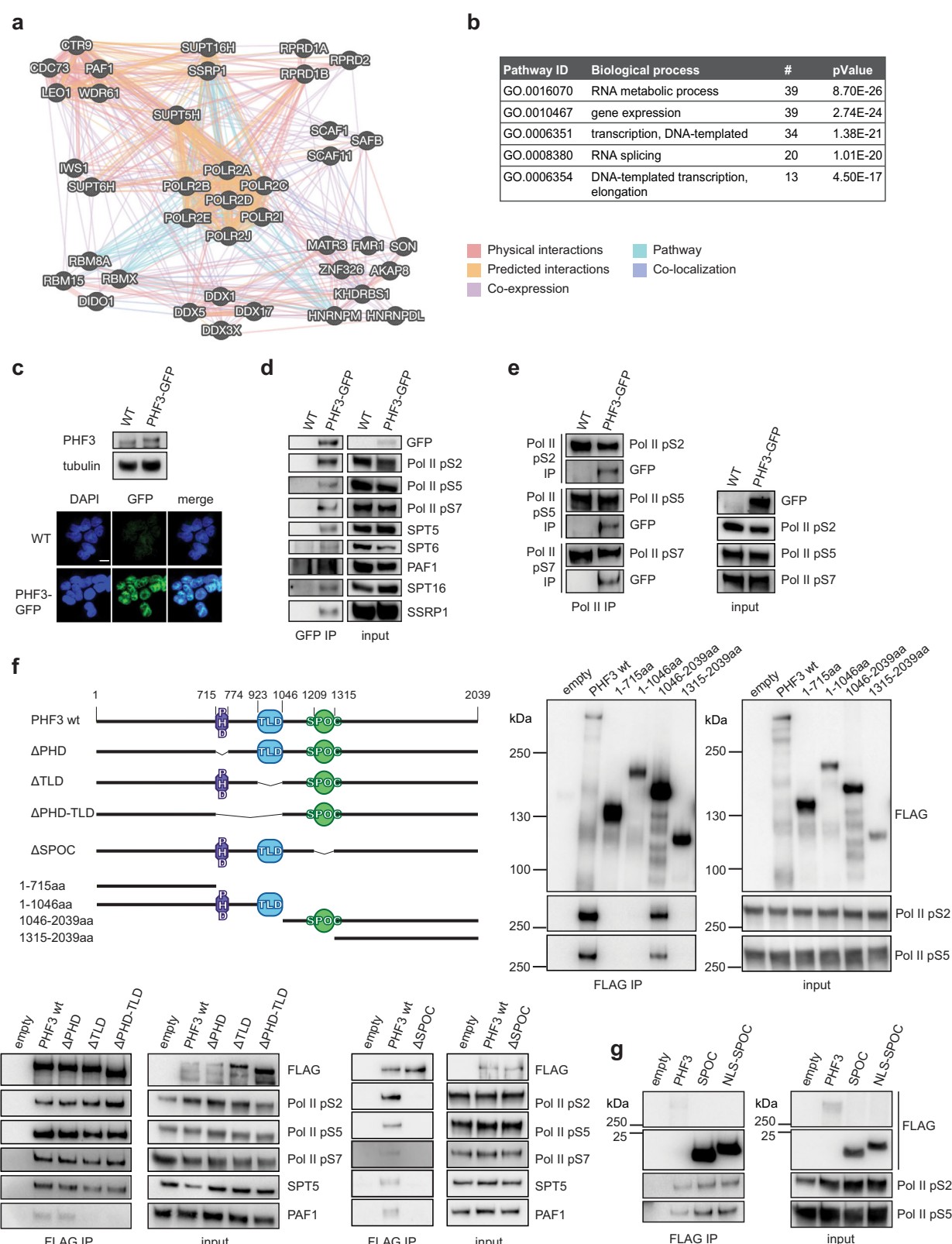

| Pathway ID | Biological process | # | pValue |
|---|---|---|---|
| GO.0016070 | RNA metabolic process | 39 | 8.70E-26 |
| GO.0010467 | gene expression | 39 | 2.74E-24 |
| GO.0006351 | transcription, DNA-templated | 34 | 1.38E-21 |
| GO.0008380 | RNA splicing | 20 | 1.01E-20 |
| GO.0006354 | DNA-templated transcription, elongation | 13 | 4.50E-17 |

that concurrent phosphorylations are more frequent along adjacent heptad repeats than within the same repeat[42,43].

To uncover the basis of the CTD heptapeptide-SPOC interaction, we determined the structure of PHF3 SPOC in the apo form (2.6 Å), and bound to 2xpS2 (2.0 Å), 2xpS2pS7 (1.75 Å) and 2xpS2pS5 (2.85 Å) diheptapeptides (Fig. 2d–h, Supplementary Fig. 2i–m, Supplementary Table 2). The three different

2xpS2-containing peptides with comparable binding affinities were used for structural analysis to determine whether PHF3 SPOC exclusively binds tandem pS2 marks or whether it can concomitantly engage pS7 or pS5 as shown for yeast Rtt103 (binds pS2pS7) and human PIN1 (binds pS2pS5)[44,45]. Superimposition of PHF3 SPOC with SPOC domains from SHARP[38] and FPA[46] revealed that PHF3 SPOC has a distorted β-barrel fold

**Fig. 1 PHF3 interacts with RNA polymerase II via the SPOC domain. a** GeneMANIA[135] interaction map of the PHF3 interactome. **b** Gene ontology biological processes of the PHF3 interactome revealed by mass spectrometry. **c** Expression levels and nuclear localization of PHF3-GFP in the endogenously tagged HEK293T cell line revealed by Western blotting with anti-PHF3 and fluorescence microscopy. Scale bar = 10 μm. The experiment was performed once. **d** PHF3-GFP was immunoprecipitated using anti-GFP. Pol II phosphoisoforms, as well as transcription regulators SPT5, SPT6, PAF1, and FACT complex (SPT16 and SSRP1) were detected in the eluates. The experiment was performed once. **e** Endogenous Pol II phosphoisoforms were immunoprecipitated from PHF3-GFP cells and PHF3-GFP was detected in the eluates. The experiment was performed once. **f** Anti-FLAG immunoprecipitation of FLAG-PHF3 deletion mutants. Pol II does not co-immunoprecipitate in the absence of the PHF3 SPOC domain. The experiment was performed four times. Representative blots are shown. **g** Anti-FLAG immunoprecipitation of the FLAG-SPOC domain shows interaction with Pol II. The experiment was performed once.

comparable to other structurally characterized SPOCs, with seven β-strands and four helices connected with various loop regions (Fig. 2d). The 2xpS2- and 2xpS2pS7-diheptapeptides were bound between strands of the β-barrel and along the α1 helix, in an extended conformation with *trans* isomer configuration of prolines (Fig. 2e–h). The two phospho groups at S2 of the diheptapeptides were electrostatically anchored by two positively charged patches on the SPOC surface, which flank the central hydrophobic patch (Fig. 2e–h and Supplementary Fig. 2i–m). SPOC binding to phosphorylated serines in adjacent repeats imposed an extended conformation on the CTD diheptapeptide (Fig. 2e–h).

The ε-amino groups K1267 and K1309 from the first patch on SPOC form a hydrogen bond with O1P from the N-terminal $pS2_a$, whereas guanidinium nitrogens from R1248 and R1297 from the second patch form hydrogen bonds with O1P and O3P from the C-terminal $pS2_b$ (Fig. 2g, h). Consistent with these findings, the R1248A substitution within SPOC reduced the binding to diheptapeptides containing pS2 more than 10-fold in vitro and reduced the interaction between PHF3 and Pol II in cells (Fig. 2b, c and Supplementary Fig. 2b–g). pS7 within the 2xpS2pS7 diheptapeptide did not form contacts with SPOC, as S7 was projected away from the SPOC surface (Fig. 2h and Supplementary Fig. 2i). I1249 and V1300 from the hydrophobic patch of SPOC form contacts with $Y1_b$, and T1253, Y1257 and Y1312 form a hydrophobic pocket for $P6_a$ (Fig. 2g, h). Hydrophobic contacts with $Y1_b$ and $P6_a$ are critical for establishing the register of the CTD to ensure specific anchoring of pS2 marks in the basic patches. All of these residues within PHF3 SPOC are generally conserved across species, as well as with DIDO SPOC (Fig. 2a, i and Supplementary Fig. 2h). Fluorescence correlation spectroscopy (FCS) showed a 1:1 stoichiometry of the SPOC:pS-diheptapeptide interaction (Supplementary Fig. 3a–c). Mass photometry showed that full length PHF3 forms monomers in vitro (Supplementary Fig. 3d), suggesting that one PHF3 molecule engages two pS2 CTD repeats through its SPOC domain. Taken together, our data show that the PHF3 SPOC domain is a previously unrecognized Pol II CTD-binding domain that preferentially recognizes the elongating form of Pol II phosphorylated on S2.

**PHF3 drives liquid-liquid phase separation of phosphorylated Pol II CTD.** Phosphorylated CTD was shown to localize inside phase-separated condensates formed by splicing factors, which bind pS2 CTD[30]. Thus we tested whether PHF3 and its SPOC domain can modulate LLPS of phosphorylated Pol II CTD. CTD was phosphorylated with the DYRK1A kinase, which preferentially targets S2[47] (Supplementary Fig. 4a). Unphosphorylated mEGFP-CTD formed condensates (median area 2.4 μm$^2$) at physiological salt concentration (150 mM NaCl), which were abrogated by 1,6-hexanediol (Fig. 3a, b and Supplementary Fig. 4b–d), according to previous findings[29]. CTD phosphorylation impaired condensate formation due to

electrostatic repulsion (median area 1.6 μm$^2$) (Fig. 3a, b and Supplementary Fig. 4b–d).

The SPOC domain alone did not form condensates, but colocalized within pS2 CTD condensates (Fig. 3a, b, Supplementary Fig. 4b–f). Unlike the isolated SPOC domain, full length PHF3 formed condensates (median area 5.2 μm$^2$) that were abrogated in the presence of 1,6-hexanediol, which interferes with hydrophobic interactions (Fig. 3a, b and Supplementary Fig. 5a–c). PHF3 condensates were larger at a higher salt concentration, confirming that PHF3 LLPS is primarily driven by hydrophobic interactions (Supplementary Fig. 5a–c). Phosphorylated CTD partitioned into PHF3 condensates and modulated their size in a salt-dependent manner (Fig. 3a, b and Supplementary Fig. 5a–c). PHF3-phCTD condensate size decreased with higher salt concentration due to interference with electrostatic interactions between PHF3 SPOC and phCTD (Supplementary Fig. 5a–c). Our results suggest that PHF3 condensates are primarily formed through hydrophobic interactions, while PHF3-phCTD condensation is additionally modulated through electrostatic interactions.

Finally, we tested whether PHF3 can drive LLPS of the Pol II complex. DYRK1A-phosphorylated Pol II (Supplementary Fig. 5d) did not undergo LLPS but partitioned into PHF3 condensates (Fig. 3a, b). PHF3-phPol II condensates were five times larger than PHF3 condensates (median area 27.8 μm$^2$), suggesting that multivalent interactions between PHF3 and phPol II promote the assembly of large condensates. PHF3 showed highest mobility within PHF3-phPol II condensates based on fluorescence recovery after photobleaching (FRAP), indicating that PHF3-phPol II interaction promotes PHF3 diffusion (Supplementary Fig. 5e, f).

**PHF3 colocalizes in Pol II clusters and travels with Pol II across the length of genes.** The physiological relevance of Pol II CTD LLPS is to drive Pol II nuclear clustering as a means of compartmentalizing transcription from RNA processing[29,30]. Super-resolution imaging revealed that PHF3 colocalizes within Pol II clusters in cells (Fig. 3c, d). To examine genome-wide colocalization between PHF3 and Pol II, we performed ChIP-seq of PHF3-GFP, Pol II pS2, Pol II pS5 and Pol II pS7. ChIP-seq analysis showed that PHF3 tracked with Pol II across the length of genes, with increasing strength from TSS (transcription start site) to pA (polyadenylation sites) (Fig. 3e and Supplementary Data 2). Additionally, higher PHF3 occupancy coincided with higher Pol II occupancy, particularly along the gene bodies (Supplementary Fig. 6), and with higher transcription levels according to single base-resolution, precision nuclear run-on and sequencing (PRO-seq) (Fig. 3f). Overall, these data indicate that PHF3 travels with the Pol II transcription machinery.

**PHF3 regulates gene expression.** To address the functional relevance of PHF3 colocalization with Pol II, we used the uridine analog EU (5-ethynyl uridine) coupled to Alexa Fluor to detect

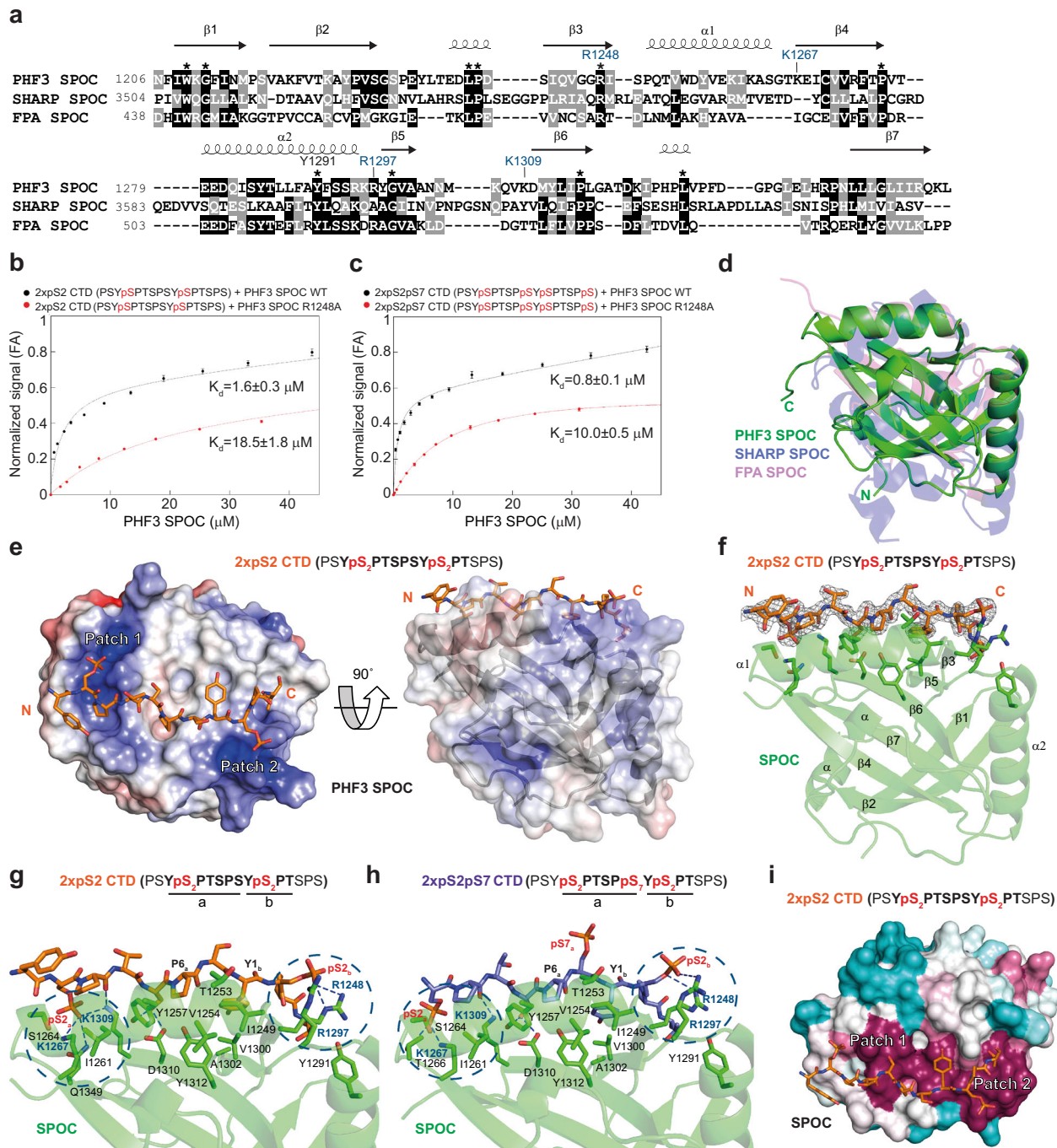

**Fig. 2 PHF3 SPOC binds pS2 CTD peptides in vitro. a** Structure-based alignment of SPOC domains from PHF3 (6Q2V), SHARP (2RT5), and FPA (5KXF). Conserved residues are marked with an asterisk. **b**, **c** Fluorescence anisotropy (FA) measurement of the binding of **b**, 2xpS2 and **c**, 2xpS2pS7 FAM-labeled CTD peptides to PHF3 SPOC WT or R1248A mutant. Normalized fluorescence anisotropy is plotted as a function of protein concentration ($n = 3$). The data were normalized for visualization purposes and the experimental isotherms were fitted to one site saturation with non-specific binding model. **d** Overlay of SPOC structures from PHF3 (6Q2V), SHARP (2RT5) and FPA (5KXF) showed an average RMSD of 2.75 Å over 149 aligned Cα atoms between PHF3 and SHARP SPOC, and average RMSD of 1.94 Å over 123 aligned Cα atoms between PHF3 and FPA SPOC. **e** 2xpS2 CTD peptide binds two positively charged patches (Patch 1 and 2) on the surface of PHF3 SPOC. The color coded electrostatic surface potential of SPOC was drawn using the Adaptive Poisson-Boltzmann Solver package within PyMol. The electrostatic potential ranges from −5 (red) to +5 (blue) kT/e. The N- and C-termini of the peptide are indicated and always shown in the same orientation. **f** 2 $F_o - F_c$ electron density map of pS2 peptide contoured at the 1.5σ level. CTD peptide sequences used for X-ray structures correspond to those used in binding assays. The residues of the CTD diheptapeptide that are visible in the structure are indicated in bold. CTD peptides used for X-ray structures had the same sequence as for the binding assays but were not fluorescently labeled. **g**, **h** Hydrogen bonding interactions between **g**, 2xpS2 and **h**, 2xpS2pS7 CTD peptides and PHF3 SPOC. SPOC monomer binds two phosphorylated S2 groups on the CTD peptides. SPOC residue labels from two positively charged patches are colored blue and the patches are contoured with dashed circles. **i** Evolutionary conservation of PHF3 SPOC residues projected onto the 2xpS2 co-structure using the ConSurf server. Residues are colored by their conservation grades with maroon showing the highest and turquoise the lowest degree of conservation. Two positively charged patches (Patch 1 and 2) are indicated.

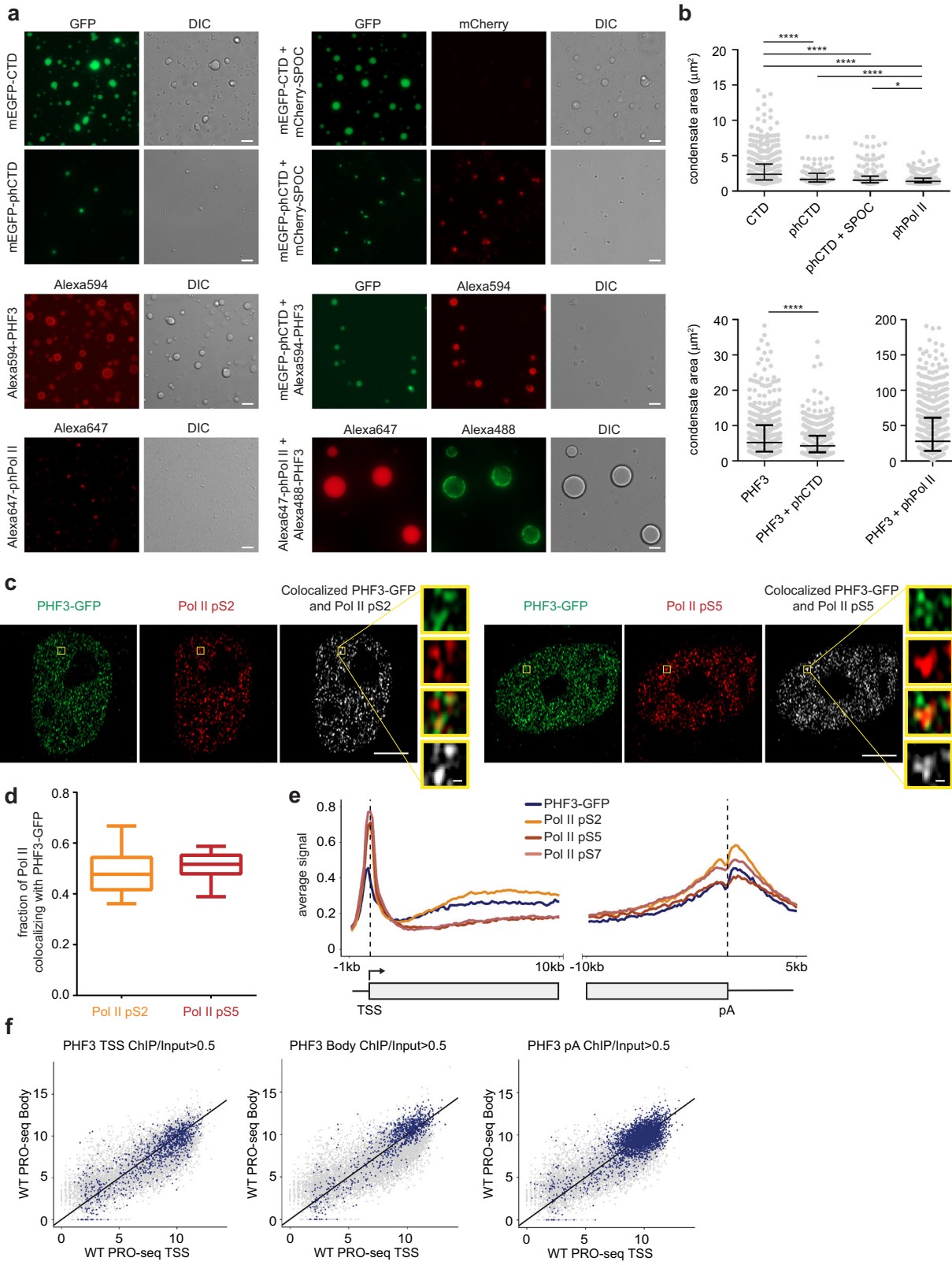

newly synthesized RNA. Super-resolution imaging revealed reduced signal in Pol II clusters that overlap with PHF3 (Fig. 4a, b), indicating that PHF3 may be associated with reduced transcriptional activity. To further elucidate the function of PHF3 in transcription, we used CRISPR/Cas9 to generate PHF3 knock-out HEK293T cells (PHF3 KO) and HEK293T cells lacking the SPOC domain (PHF3 ΔSPOC) (Supplementary Fig. 7). PHF3 ΔSPOC was expressed at lower levels

compared to PHF3 WT suggesting that removal of the SPOC domain may reduce protein stability (Supplementary Fig. 7c). We initially assessed the effect of PHF3 loss or SPOC deletion by monitoring incorporation of EU during transcription by confocal microscopy or FACS (Fig. 4c–e and Supplementary Fig. 8). PHF3 KO and PHF3 ΔSPOC cells showed significantly higher EU levels than WT cells (Fig. 4c–e), suggesting increased transcript levels.

**Fig. 3 PHF3 drives liquid-liquid phase separation of phosphorylated Pol II, colocalizes in Pol II clusters in cells and associates with Pol II genome-wide.**
**a** Representative images of in vitro LLPS assays with 5 μM unphosphorylated or phosphorylated mEGFP-CTD, 5 μM mCherry-SPOC, 3 μM Alexa594-PHF3, 3 μM phosphorylated Alexa488-Pol II, 1.5 μM phosphorylated Alexa647-Pol II + 1.5 μM Alexa488-PHF3. Scale bar = 5 μm. The experiments were repeated three times and the representative images are shown. **b** Quantification of condensate area (μm$^2$). N = 556 (CTD); 64 (phCTD); 89 (phCTD+SPOC); 582 (phPol II); 480 (PHF3); 580 (PHF3+phCTD); 588 (PHF3+phPol II). Data are presented as median with interquartile range. Mann-Whitney test was used to determine p-values (Supplementary Data 7). **c** Representative Airyscan high resolution images of PHF3-GFP (IF staining with rabbit anti-GFP + Alexa Fluor 488, green) and Pol II pS2 or pS5 (Alexa Fluor 594, red). Co-localization analysis of clusters that overlap in both channels (white). Scale bar = 5 μm or 200 nm for zoomed regions. **d** Quantification of the fraction of Pol II pS2 (N = 28) and Pol II pS5 (N = 23) co-localizing with PHF3. Box and whiskers plot with error bars representing 10 and 90 percentiles are shown. Each experiment was repeated three times with comparable results. Statistics are indicated in detail in Supplementary Data 7. **e** ChIP-seq analysis shows that PHF3 travels with Pol II across the length of genes. Relative enrichment of PHF3, Pol II pS2, pS5, and pS7 on TSS-gene body region (TSS viewpoint; left panel) and gene body-pA region (pA viewpoint; right panel) for genes that showed Pol II occupancy with the F-12 antibody (minimal gene body RPKM of 5 in F-12 ChIP-seq). **f** Scatter plots showing PRO-seq nascent transcription levels at gene body relative to TSS in WT cells. Blue dots indicate PHF3-bound genes at transcription start sites (TSS, left), gene body (Body, middle) or polyadenylation sites (pA, right).

To investigate the changes in RNA levels genome-wide, we performed RNA-seq to measure mature transcripts, and PRO-seq to measure nascent transcripts (Fig. 4f–j, Supplementary Figs. 9, 10 and Supplementary Data 3). We found that mature transcript levels were generally increased in both PHF3 KO and ΔSPOC cells compared to WT: 620 and 638 mature transcripts were elevated >2-fold (p-value<0.05) in PHF3 KO or ΔSPOC cells relative to WT, with 281 (~45%) elevated in both conditions (Fig. 4f and Supplementary Fig. 9a). In contrast, only 173 and 74 mature transcripts were downregulated >2-fold in PHF3 KO or ΔSPOC relative to WT, with 37 downregulated in both conditions.

Unlike mature transcripts, nascent transcripts did not show major changes in PHF3 KO and ΔSPOC cells (Fig. 4g and Supplementary Fig. 9b). We observed increased nascent transcription of 68 and 78 genes >2-fold in PHF3 KO or ΔSPOC cells relative to WT, with 29 elevated in both conditions. Similarly, 39 and 70 genes showed decreased transcription >2-fold in PHF3 KO or ΔSPOC cells relative to WT, with 10 reduced in both conditions. About 10% of the mature transcripts with elevated steady-state levels in PHF3 KO or ΔSPOC cells had a concomitant increase in nascent transcripts (orange dots in Fig. 4h; example of an RNA-seq/PRO-seq upregulated gene in Fig. 4i; example of an RNA-seq upregulated gene in Fig. 4j). Overall, these data suggest that loss of PHF3 causes a derepression of a subset of genes, and that the binding of PHF3 to Pol II via the SPOC domain contributes to its function.

**PHF3 loss results in increased Pol II stalling and reduced elongation rate.** To understand how PHF3 may regulate Pol II transcription, we analyzed Pol II distribution in PHF3 WT, KO, and ΔSPOC by PRO-seq and Pol II ChIP-seq (Fig. 5a, b and Supplementary Fig. 9c, d). Metagene profiles showed a small reduction in PRO-seq nascent transcript levels in the gene bodies and towards polyadenylation (pA) sites ('All genes' in Fig. 5a and Supplementary Fig. 9c). Pol II occupancy was elevated at TSS and reduced along gene bodies in PHF3 KO and ΔSPOC cells relative to WT cells ('All genes' in Fig. 5a). A change in Pol II occupancy may indicate impaired pause release[5,13,14,48]. Pol II stalling index (TSS reads/gene body reads)[14] showed increased stalling in PHF3 KO cells based on Pol II occupancy but no difference based on PRO-seq ('All genes' in Fig. 5b). The same type of analysis for RNA-seq upregulated genes revealed increased stalling in PHF3 KO and ΔSPOC cells based on both PRO-seq and Pol II occupancy ('RNA-seq UP genes' in Fig. 5a, b).

Considering that a change in Pol II distribution was also linked with a change in elongation rate and that elongation factors RECQL5, PAF1, and SCAF8 were shown to modulate elongation rate[11,15,16,49], we measured in vivo transcription elongation rate

by blocking pause release with a CDK9 inhibitor 5,6-dichloro-1-β-D-ribofuranosyl-benzimidazole (DRB) applied for 3.5 h, followed by DRB washout to allow transcription for 10, 25, and 40 min (Fig. 5c). PRO-seq analysis of the front edge of waves of nascent transcription allowed us to deduce Pol II elongation rate (Supplementary Data 4). PHF3 loss or SPOC deletion led to a reduction in elongation rate after 10 min (median elongation rate: 2.62 kb/min for KO, 2.66 kb/min for ΔSPOC and 3.37 kb/min for WT) followed by a subsequent increase and leveling off by the 40 min timepoint, without having major effects on Pol II processivity (Fig. 5d, e). Genes with reduced elongation rate consistently showed a decrease in nascent transcript levels revealed by PRO-seq analysis (Fig. 5f and Supplementary Fig. 11a). Interestingly, genes with reduced elongation rate also showed a mild tendency towards upregulation in RNA-seq in PHF3 KO and ΔSPOC (fold change<2), indicating coupling between reduced elongation and an increase in mature transcript levels (Fig. 5g and Supplementary Fig. 11b). Taken together, our data show that PHF3 globally promotes pause release and Pol II elongation. However, its loss has little effect on nascent RNA levels but causes an increase in mRNA levels.

**PHF3 loss results in increased mRNA stability.** RNA-seq and PRO-seq analyses revealed that a subset of ~600 genes showed an increase in mature transcripts without major alteration in nascent transcription levels, indicating that PHF3 loss leads to increased mRNA stability. We calculated the difference in log2 fold changes KO/WT or ΔSPOC/WT between RNA-seq as a measure of steady-state mRNA content, and PRO-seq as a measure of mRNA production rate (Fig. 6a). Differences were strikingly skewed towards positive values, indicating that the changes in the steady-state mRNA levels are greater than the changes in transcriptional rates (Fig. 6a). This finding strongly suggested that PHF3 regulates mRNA stability. To validate this further, we determined mRNA half-lives by performing SLAM-seq [Thiol (SH)-Linked Alkylation for the Metabolic sequencing of RNA][50] (Supplementary Fig. 12a). Cells were pulse-labeled for 12 h with s4U (0 h sample), followed by a chase with uridine for 6 and 12 h. The T-C conversion rate was used to determine mRNA half-lives for genes that showed conversion rates >0 at the 0 h timepoint and a monotonic decrease in median conversion rates (Fig. 6b, Supplementary Fig. 12 and Supplementary Data 5).

We observed a significant overall increase in mRNA half-lives in PHF3 KO and ΔSPOC compared to WT (median half-lives: 5.948694 h for PHF3 KO, 5.970330 h for ΔSPOC and 4.837077 h for WT; Fig. 6b). 62% genes showed increased mRNA half-lives in both PHF3 KO and ΔSPOC (Fig. 6c), and an additional 27% showed an increased half-life in either KO or ΔSPOC (Supplementary Data 5). Because global SLAM-seq analysis failed

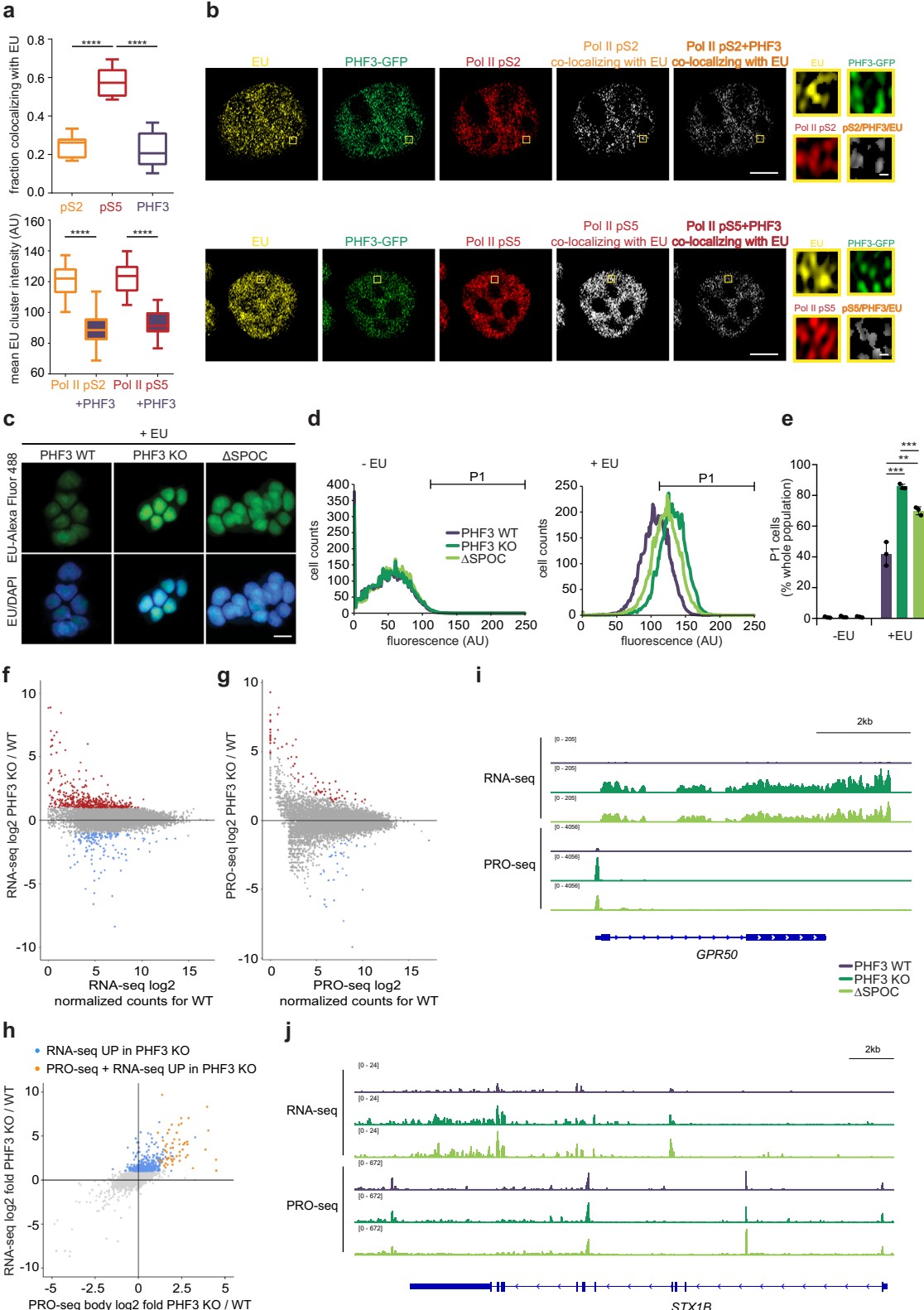

to capture low-expressed genes, which exhibit the strongest upregulation in PHF3 KO and ΔSPOC (Supplementary Fig. 12d), we applied a targeted SLAM-seq approach to specifically analyze mRNA half-lives for *INA* as one of the most highly upregulated genes in PHF3 KO and ΔSPOC. *INA* showed a pronounced increase in mRNA half-life from 3.3 h in WT to 7.1 h in ΔSPOC and 19.5 h in PHF3 KO (Fig. 6d), which was not observed for the housekeeping gene *NAT10* (Fig. 6e). Comparison of changes in RNA-seq and SLAM-seq showed that the increase in mRNA stability was concordant with the increase in mature transcript levels (Fig. 6f, g). Moreover, increased mRNA stability in PHF3 KO/ΔSPOC may be coupled with reduced elongation rate, as genes with reduced elongation rate also had longer mRNA half-lives in PHF3 KO/ΔSPOC (Fig. 6h, i).

**Fig. 4 PHF3 negatively regulates mRNA levels. a** High-resolution Airyscan imaging reveals a high degree of co-localization between Pol II pS5 and 5-EU whereas only a small fraction of Pol II pS2 or PHF3-GFP co-localizes with 5-EU (N = 21 for pS5; N = 15 for pS2; N = 23 for PHF3). The mean EU intensity is decreased in clusters where PHF3 and Pol II overlap compared to clusters containing only Pol II. Box and whiskers plots with error bars representing the 10 and 90 percentiles are shown. One-way ANOVA with Tukey's multiple comparison test was performed to determine p-values (<0.0001). Each experiment was repeated twice with comparable results. Statistics are indicated in detail in Supplementary Data 7. **b** Representative Airyscan high resolution images of 5-EU (yellow), PHF3-GFP (green) and Pol II pS2 or pS5 (red) and clusters of Pol II that co-localize with EU or with both EU and PHF3 (white). Scale bar = 5 μm or 200 nm for zoomed regions. **c–e** PHF3 KO and ΔSPOC show increased incorporation of EU-Alexa Fluor 488 by **c**, fluorescence microscopy (scale bar = 10 μm) and **d**, **e** FACS analysis. **d** Cell counts for each fluorescence intensity in the absence (-EU) or presence of EU (+EU). **e** Percentage of cells belonging to the gated P1 fluorescent population shown in **d**. The following cell numbers were examined over three independent experiments: N = 29059 (WT −EU), N = 29918 (KO −EU), N = 29847 (ΔSPOC −EU), N = 29146 (WT +EU), N = 29649 (KO +EU), N = 29771 (ΔSPOC +EU). Data are presented as mean values ± standard deviation. One-tailed, two-sample equal variance t-test was used to determine p-values (Supplementary Data 7). **f** RNA-seq analysis shows upregulation of 620 genes (red dots, fold-change>2, p < 0.05) and downregulation of 173 genes (blue dots, fold-change>2, p < 0.05) in PHF3 KO compared to WT. Drosophila S2 cells were used for spike-in normalization. **g** PRO-seq analysis shows upregulation of 68 genes (red dots, fold-change>2, p < 0.05) and downregulation of 39 genes (blue dots, fold-change>2, p < 0.05) in PHF3 KO compared to WT. Drosophila S2 nuclei were used for spike-in normalization. Mean Pearson correlation coefficient between the samples was 0.96 (see Supplementary Fig. S10c). **h** Relationship between RNA-seq and PRO-seq fold change for PHF3 KO vs WT. Genes that are upregulated in PHF3 KO in RNA-seq but not PRO-seq are indicated in blue. Genes that are upregulated in both RNA-seq and PRO-seq are indicated in orange. **i** Integrative Genomics Viewer (IGV) snapshots showing RNA-seq and PRO-seq reads for *GPR50* as a typical gene with increased RNA-seq and PRO-seq in PHF3 KO and ΔSPOC cells. **j** IGV snapshots showing RNA-seq and PRO-seq reads for *STX1B* as a typical gene with increased RNA-seq but no change in PRO-seq in PHF3 KO and ΔSPOC cells. RNA-seq was performed after Ribo-Zero treatment of total RNA.

Overall, our data define PHF3 as an elongation factor that regulates different aspects of RNA biogenesis, spanning from transcription to RNA metabolism.

**PHF3 negatively regulates a small subset of genes by competing with TFIIS**. The most profound changes in gene expression following PHF3 or SPOC depletion affected a small subset of ~70 genes, which were highly upregulated at both the nascent and mature transcript level (RNA-seq/PRO-seq upregulated, Fig. 4h). Genes with highly elevated transcripts in PHF3 KO and ΔSPOC are transcribed at low levels in WT cells and enriched for both the repressive H3K27me3 mark and the active H3K4me3, as well as pS5 (Fig. 7a, b, Supplementary Fig. 13, Supplementary Data 2). ChIP analysis showed a decrease in H3K27me3 on these genes in PHF3 KO and ΔSPOC cells (Fig. 7c). Accordingly, super-resolution imaging showed reduced colocalization of Pol II with H3K27me3 in PHF3 KO and ΔSPOC (Fig. 7d, e), suggesting that PHF3-mediated transcriptional derepression is coupled with the loss of Polycomb-mediated silencing. Overall, these data suggest that Pol II is in a poised state and possibly undergoing fast turnover at this group of genes in WT cells[51,52].

Poised Pol II may experience high levels of backtracking due to high GC content, specific promoter elements, or chromatin configuration[3,53]. Backtracked polymerase is rescued by the positive elongation factor TFIIS, which stimulates cleavage of the nascent RNA and allows Pol II to resume transcription[17,35,54,55]. Given that PHF3 harbors a TFIIS-like domain (TLD), which could potentially displace TFIIS from Pol II, we examined TFIIS binding profile in PHF3 WT, KO, and ΔSPOC cells by ChIP. TFIIS occupancy was increased most prominently on genes that were both PRO-seq and RNA-seq upregulated in PHF3 KO or ΔSPOC and did not change on genes that were not transcriptionally deregulated (Fig. 7f, g and Supplementary Data 2). TFIIS and the TLD domain of the yeast PHF3 homolog Bye1 bind a similar region on Pol II[31] (Supplementary Fig. 14). To test whether PHF3 can compete with TFIIS for binding to Pol II, we used sucrose gradient ultracentrifugation to analyze complexes formed between Pol II-EC (elongation complex) and PHF3 in the presence of TFIIS as a competitor, and vice versa (Supplementary Fig. 15a). We used an inactive TFIIS mutant (TFIIS^M; D282A E283A) to prevent RNA cleavage and Pol II-EC disassembly. Although TFIIS could not displace PHF3 from Pol II-EC, PHF3 almost completely displaced TFIIS from Pol II-EC (Supplementary Fig. 15a). These data suggest

that PHF3 competes with TFIIS for binding to Pol II in vitro. TFIIF, which cooperates with TFIIS to rescue arrested Pol II[56,57], stabilized the association between TFIIS and Pol II but did not prevent PHF3-mediated displacement of TFIIS (Supplementary Fig. 15b). TFIIF itself was not displaced by PHF3, nor did it outcompete PHF3 from the Pol II-EC (Supplementary Fig. 15b), indicating that PHF3 does not compete with TFIIF. Thus, even in the presence of TFIIF, PHF3 outcompetes TFIIS for binding to Pol II. Furthermore, we tested whether PHF3 inhibits the function of wild-type TFIIS in a transcription assay using an arrest template. Pol II-EC efficiently transcribed the arrest sequence in the presence of TFIIS, but elongation was diminished in the presence of PHF3 (Fig. 7h). PHF3 lacking the TLD domain (PHF3 ΔTLD) did not interfere with TFIIS-dependent elongation, confirming that the PHF3 TLD domain is critical for competition with TFIIS (Supplementary Fig. 15c). PHF3 TLD alone did not affect TFIIS-dependent elongation, as the SPOC domain is required for docking PHF3 onto Pol II (Supplementary Fig. 15d). Taken together, our results suggest that PHF3 negatively regulates transcription of a small subset of genes by competing with the transcription factor TFIIS and impairing rescue of backtracked Pol II. PHF3 competition with TFIIS explains the increase in both nascent and mature transcripts observed for a subset of genes in the absence of PHF3 or its SPOC domain.

**PHF3 regulates neuronal gene expression and is required for neuronal differentiation of mESCs**. GO analysis of the upregulated transcripts in PHF3 KO and ΔSPOC HEK293T cells revealed an enrichment of neuronal genes (Fig. 8a and Supplementary Fig. 9e, f), whereas downregulated or unaffected genes did not show any particular functional enrichment. To consolidate the tissue specificity of transcriptionally upregulated genes, we used the TissueEnrich tool[58], which revealed cerebral cortex as the only tissue with significant enrichment (Fig. 8b and Supplementary Fig. 9g).

The neuronal genes *INA* and *GPR50* exhibited low mRNA levels in WT HEK293T cells, whereas loss of PHF3 or the PHF3 SPOC domain resulted in a pronounced increase in their mRNA levels and increased occupancy of Pol II throughout the gene (Fig. 8c and Supplementary Fig. 16). Exogenous expression of PHF3 in PHF3 KO cells reduced the expression of *INA* and *GPR50* mRNAs to levels that were comparable to WT cells

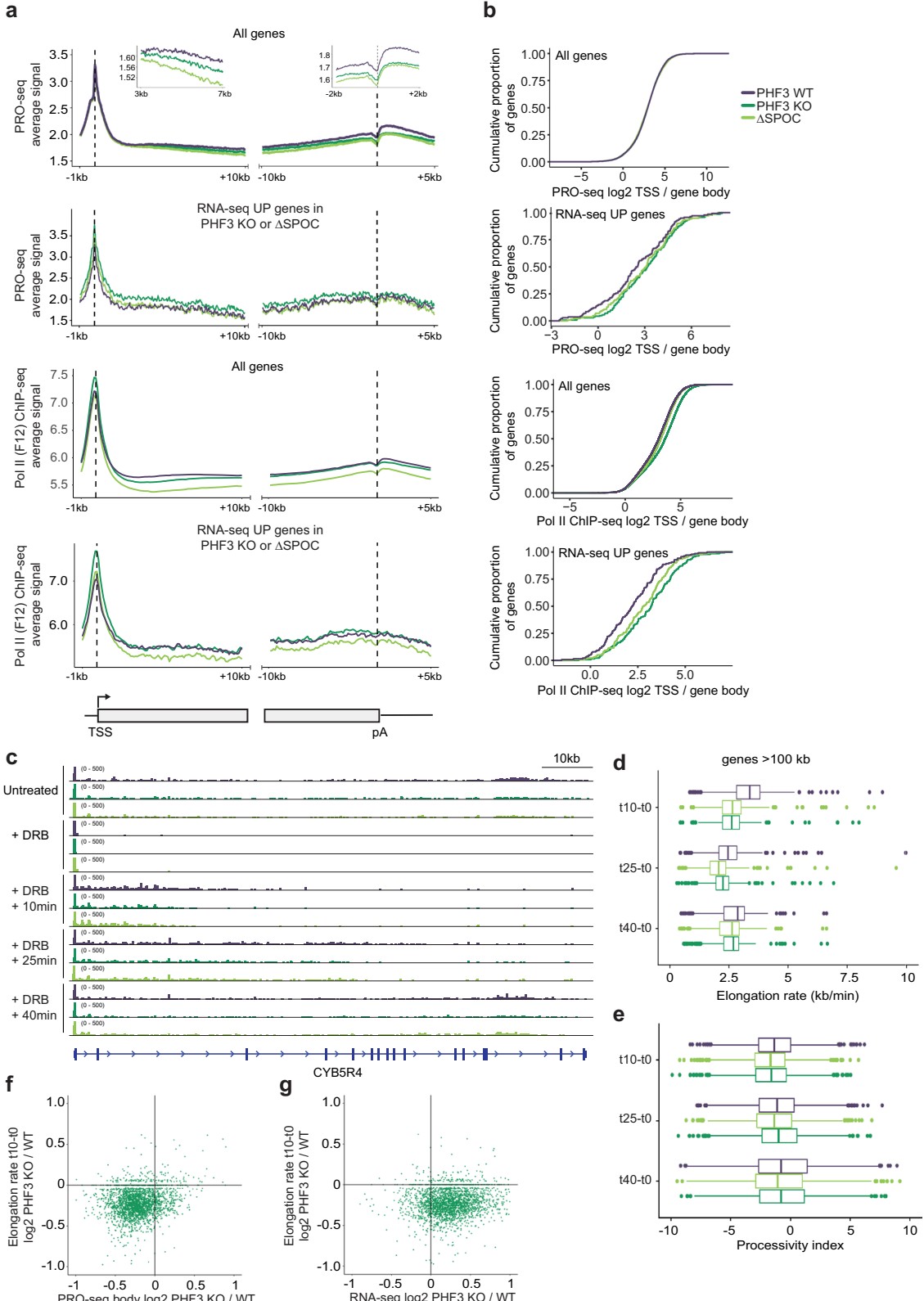

(Fig. 8c). Importantly, PHF3 ΔSPOC failed to rescue the PHF3 KO phenotype (Fig. 8c).

Many of the neuronal genes that we found to be directly repressed by PHF3, including *INA*[59] and *GPR50*[60], have been implicated in different aspects of neurodevelopment. Moreover, PHF3 expression levels were found to be reduced in glioblastoma, the most common undifferentiated brain tumor[61]. To examine a

potential role for PHF3 in proper neuronal differentiation, we used CRISPR/Cas9 to generate Phf3 KO mouse embryonic stem cells (mESCs) (Fig. 8d and Supplementary Figs. 17 and 18). We differentiated Phf3 KO and WT mESCs into neural stem cells (NSCs), and subsequently into neurons or astrocytes. As expected, NSCs derived from WT mESCs could differentiate into beta III tubulin (TuJ1)-positive neurons organized in extensively

**Fig. 5 PHF3 loss results in increased Pol II stalling and reduced elongation rate. a** Composite analysis of PRO-seq and Pol II (F-12) ChIP-seq distribution and signal strength in PHF3 WT, KO, and ΔSPOC on TSS-gene body region and gene body-pA region for all genes or RNA-seq upregulated genes (fold-change>2, $p < 0.05$). Mouse chromatin was used for spike-in normalization of ChIP-seq. **b** Stalling index analysis calculated as PRO-seq or Pol II ChIP-seq TSS/gene body signal for all genes or RNA-seq upregulated genes (fold-change>2, $p < 0.05$). **c** IGV snapshots showing PRO-seq reads for CYB5R4 for the elongation rate experiment. Pause release was blocked with the CDK9 inhibitor DRB for 3.5 h, followed by DRB washout to allow transcription for 10, 25, and 40 min. **d** Pol II elongation rate for genes >100 kb ($N = 795$) was calculated as the leading edge of waves of nascent transcription divided by the time after DRB washout. Data are presented as box and whiskers plots showing the median and the interquartile range. **e** Processivity index for genes >100 kb ($N = 795$) was calculated as log10 distal/proximal reads. Data are presented as box and whiskers plots showing the median and the interquartile range. **f** Relationship between PRO-seq body fold change and t10 elongation rate (10 min after DRB washout) fold change for PHF3 KO vs WT. **g** Relationship between RNA-seq fold change and t10 elongation rate (10 min after DRB washout) fold change for PHF3 KO vs WT.

connected neuronal clumps (Fig. 8e, f). WT NSCs could also differentiate into glial fibrillary acidic protein (GFAP)-positive astrocytes (Fig. 8f). In contrast, although NSCs derived from Phf3 KO mESCs formed astrocytes comparable to WT NSCs, they failed to differentiate into properly shaped and connected neurons (Fig. 8e, f). Additionally, we found that Phf3 transcript levels were elevated in WT NSCs relative to WT mESCs, suggesting that Phf3 expression is regulated during neuronal differentiation (Fig. 8g). Although we cannot completely exclude the possibility that loss of Phf3 might affect other differentiation pathways, our findings suggest that Phf3 is required for proper terminal differentiation of NSCs, specifically into the neuronal lineage.

We hypothesized that loss of Phf3 triggers derepression of specific genes that must be tightly regulated for efficient neuronal differentiation. To test this, we analyzed Phf3 KO mESCs by RNA-seq. Indeed, loss of Phf3 led to sustained upregulation of several factors that are important for neuronal fate specification, such as Ascl1, Pou3f2, Sox21, and Nestin, in mESCs, NSCs, and neurons (Fig. 8h, Supplementary Data 6). The pioneer proneural transcription factor Ascl1 must be tightly regulated for the development and proliferation of NSCs, as well as for the differentiation of progenitors along the neuronal lineage[62]. The transcription factor Pou3f2 (also called Oct7 or Brn2) acts downstream of Ascl1 and is required for the differentiation of neural progenitor cells into functional neurons[63]. High levels of the intermediate filament Nestin are also expected to interfere with terminal neuronal differentiation[64]. Upregulation of the transcription factor Sox21 induces premature expression of neuronal markers but also inhibits terminal neuronal differentiation[65–67]. Additionally, the stemness marker Sox2, which promotes neural stemness specification and suppresses neuronal differentiation[68], was upregulated in Phf3 KO NSCs and neurons relative to WT controls (Supplementary Fig. 17d).

Whereas neuronal factors were upregulated upon loss of Phf3, the embryonic stemness markers Oct4 and Nanog showed reduced expression in Phf3 KO in NSCs and neurons relative to WT controls (Supplementary Fig. 17d). In line with the premature expression of neural markers, Phf3 KO mESCs showed accelerated exit from naive pluripotency compared to WT mESCs, as observed by upregulation of Pou3f1 (Oct6), Fgf5, Otx2, and Pax6 within the first 24 h of differentiation (Supplementary Fig. 17e). Taken together, our data suggest that Phf3 KO mESCs fail to differentiate into neurons due to aberrant and precocious derepression of factors that regulate neuronal commitment and terminal differentiation.

## Discussion
Here, we established PHF3 as a Pol II regulator that couples transcription with mRNA stability. In addition, we discovered that PHF3 is required for proper neuronal differentiation by preventing the precocious expression of a subset of neuronal genes. We found that PHF3 binds to the Pol II CTD phosphorylated on S2 through a CTD-binding domain called SPOC.

Our study reveals that the PHF3 SPOC domain is a phospho-serine binding module akin to SHARP SPOC. Specific recognition of phosphorylated Serine is achieved through conserved electrostatic interactions with Arginine and Lysine residues and conserved hydrophobic interactions that involve Tyrosine. PHF3 SPOC specifically docks onto phosphorylated Pol II CTD, whereas SHARP SPOC engages with the phosphorylated LSD motif of SMRT/NCoR co-repressors[38,39]. Despite the versatility in binding partners, SPOC domain proteins seem to universally regulate gene expression and differentiation. Here we showed that PHF3 regulates neuronal gene expression and neuronal differentiation, while SHARP (SPEN) was shown to regulate neural and hematopoietic differentiation through transcriptional repression of the Notch pathway[69,70] and is crucial for Xist-mediated X-chromosome silencing[71–75]. Another SPOC domain protein, RBM15, regulates N6-methyladenosine (m6A) RNA modification, splicing, and mRNA export, and inhibits myeloid differentiation[76–79]. The yeast PHF3 homolog Bye1 negatively regulates early stages of transcription elongation[32,80], while Arabidopsis SPOC proteins FPA and BORDER regulate 3′end pausing and processing[81,82]. Future characterization of SPOC domains from different proteins and species will address some key questions: is the SPOC domain a universal phospho-serine recognition module, how is the choice of binding partners determined, and how exactly do they regulate gene expression to ensure proper differentiation.

Specific recognition of phosphorylated Serine-2 of the Pol II CTD classifies PHF3 SPOC as a CTD reader domain, joining the company of (i) the CTD-interaction domain (CID) of yeast Rtt103 (pS2pS7, pT4), Pcf11 (pS2), Nrd1 (pS5), mammalian SCAF8 (pS2) and RPRD1A/B (pS2, pS7), (ii) the FCPH domain of mammalian Scp1 (pS5), (iii) the nucleotidyltransferase (NT) domain of yeast Cgt1, Pce1 and mammalian Mce1 (pS5), and (iv) the WW domain of mammalian PIN1 (pS2pS5)[22]. Despite their structural diversity, all CTD readers establish electrostatic interactions with phospho-groups and hydrophobic stacking interactions between CTD reader Tyrosine and CTD Proline. Different CTD readers can have different CTD sequence recognition requirements, which can influence CTD secondary structure[22]. Our structural analysis revealed that PHF3 SPOC requires two consecutive pS2 marks for stable binding and imposes an extended conformation of the CTD.

While the Pol II CTD is the docking point for PHF3, this large mammalian protein can likely establish additional contacts with Pol II, such as through the TLD domain, which was shown to bind the jaw-lobe domain of Pol II in the case of the yeast homolog Bye1[31] (Supplementary Fig. 14). Bivalent interaction of PHF3 with Pol II, through the Pol II jaw-lobe domain and through the CTD, may explain the dual function of PHF3 in transcription and mRNA stability.

How does PHF3 regulate transcription? By comparing mature (RNA-seq) and nascent (PRO-seq) transcript levels as well as Pol II occupancy (ChIP-seq), we identified global and gene-specific

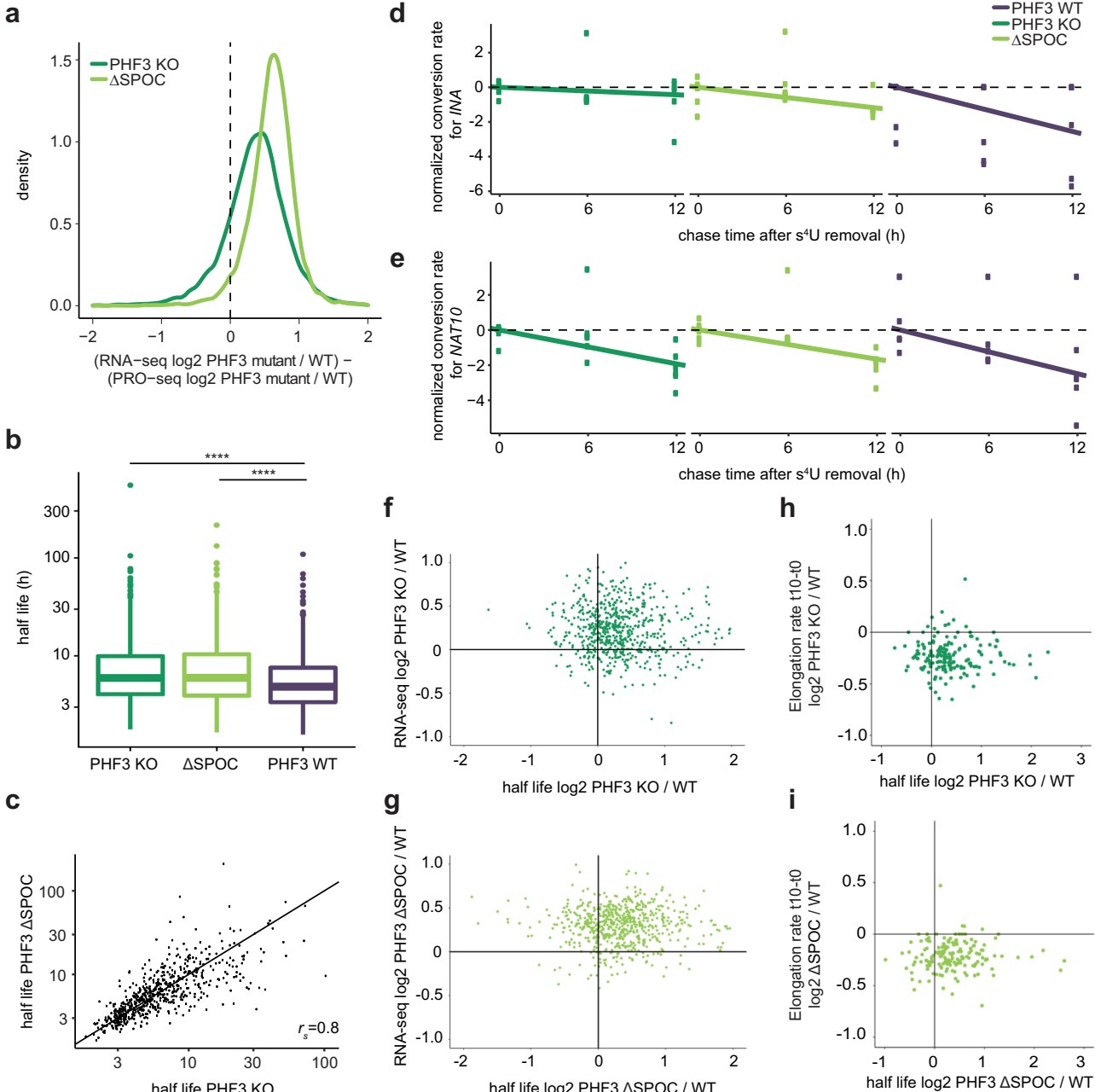

**Fig. 6 PHF3 negatively regulates mRNA stability via the SPOC domain. a** Density distribution of the differences in log2 fold changes PHF3 KO/WT or PHF3 ΔSPOC/WT between RNA-seq and PRO-seq data. **b** Comparison of mRNA half-lives for 757 genes calculated from T-C conversion rates as determined by SLAM-seq in PHF3 WT, KO, and ΔSPOC cells ($n = 6$). Median Spearman correlation coefficient of conversion rates for replicate samples belonging to the same group (same genotype and timepoint) was 0.75 (see Supplementary Fig. 12c). The difference between the distributions is statistically significant based on the one-sided Wilcoxon test [$P$(KO – WT) = $1.34 \times 10^{-11}$, $P$(ΔSPOC – WT) = $2.28 \times 10^{-11}$]. Statistics are indicated in detail in Supplementary Data 7. **c**, Scatter plot showing correlation between half-lives in PHF3 ΔSPOC and PHF3 KO. Spearman's correlation coefficient is indicated. **d**, **e** Conversion rates determined from targeted SLAM-seq analysis of **d**, *INA* mRNA and **e**, *NAT10* mRNA as a control labeled with s⁴U for 12 h followed by pulse chase for 6 h and 12 h. Robust linear models were fit on the linearized form of the exponential decay equation. Y-axis shows the log2 conversion rate, shifted by the median conversion rate at t = 0 h. For *INA*: $t_{1/2}$ = 3.3 h (WT), 7.1 h (ΔSPOC), 19.5 h (PHF3 KO). For *NAT10*: $t_{1/2}$ = 3.3 h (WT), 4.3 h (ΔSPOC), 5.0 h (PHF3 KO). **f**, **g** Relationship between RNA-seq fold change and half-life fold change for **f**, PHF3 KO vs WT or **g**, PHF3 ΔSPOC vs WT. The majority of differentially regulated genes cluster in the top right quadrant that corresponds to mRNAs with increased steady-state levels and half-lives. **h**, **i** Relationship between t10 elongation rate (10 min after DRB washout) fold change and half-life fold change for **h**, PHF3 KO vs WT or **i**, PHF3 ΔSPOC vs WT.

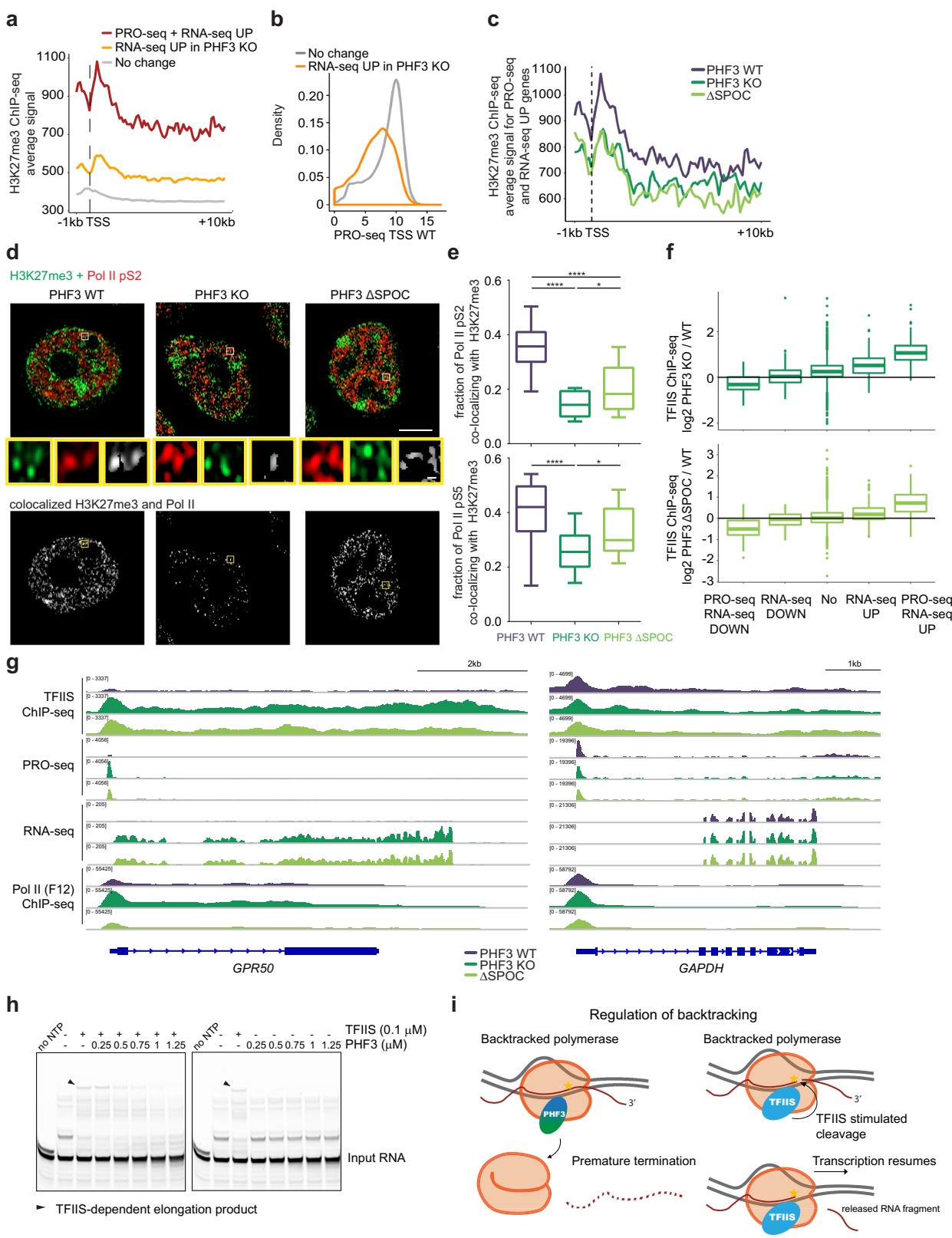

effects of PHF3 loss. Globally, PHF3 loss led to increased Pol II stalling and reduced elongation rate, with little effect on nascent transcript levels but with an increase in mRNA stability. A subset of ~600 genes was strongly derepressed (>2-fold increase in mature transcript levels), among which ~70 genes also showed a concomitant increase in nascent transcription. While globally PHF3 seems to act as a positive regulator of transcription

elongation and a negative regulator of RNA stability, on a small subset of genes PHF3 exerts negative effects on both transcription and RNA stability.

Yeast genetic studies have previously suggested competition between the yeast homolog Bye1 and Dst1 (TFIIS) for Pol II binding[32,34] but this model has never been experimentally tested. Using in vitro reconstitution, we showed that PHF3 competes

**Fig. 7 PHF3 negatively regulates a small subset of genes by competing with TFIIS. a** Composite analysis of H3K27me3 distribution and signal strength in PHF3 WT cells on TSS-gene body region for different gene categories based on RNA-seq and PRO-seq data. **b** Genes upregulated in PHF3 KO cells according to RNA-seq (fold change>2) have low expression levels in WT cells as judged by nascent transcription (PRO-seq) levels at TSS. **c** Composite analysis of H3K27me3 distribution and signal strength in PHF3 WT, KO, and ΔSPOC cells on TSS-gene body region for genes upregulated in RNA-seq and PRO-seq in PHF3 KO or ΔSPOC cells (fold change>2). **d** Representative Airyscan high resolution images of Pol II pS2 (Alexa Fluor 594, red) and H3K27me3 (Alexa Fluor 488, green) in PHF3 WT, KO, or ΔSPOC cells. Co-localization analysis of clusters that overlap in both channels (white). Scale bar = 5 μm. **e** Quantification of the fraction of Pol II pS2 and Pol II pS5 co-localizing with H3K27me3 (pS2: $N = 20$ for WT; $N = 14$ for KO; $N = 15$ for ΔSPOC. pS5: $N = 18$ for WT; $N = 22$ for KO; $N = 15$ for ΔSPOC.). Box and whiskers plots with error bars representing the 10 and 90 percentiles are shown. One-way ANOVA with Tukey's multiple comparison test was performed to determine p-values (****<0.0001; * = 0.04). Each experiment was repeated twice with comparable results. Statistics are indicated in detail in Supplementary Data 7. **f** TFIIS ChIP-seq log2 fold change PHF3 KO/WT (top) or PHF3 ΔSPOC/WT (bottom) for TSS. Genes were grouped according to changes in RNA-seq and PRO-seq: downregulation in RNA-seq (fold change>2; $N = 128$), upregulation in RNA-seq (fold change>2; $N = 395$), downregulation in RNA-seq and PRO-seq (fold change>2; $N = 25$), upregulation in RNA-seq and PRO-seq (fold change>2; $N = 45$) or no change ($N = 15868$). Statistics are indicated in detail in Supplementary Data 7. Mouse chromatin was used for spike-in normalization of TFIIS ChIP-seq. **g** IGV snapshots showing TFIIS ChIP-seq, PRO-seq, RNA-seq, and Pol II ChIP-seq (F-12) reads for GPR50 (left) and GAPDH (right) as a housekeeping gene. RNA-seq was performed after Ribo-Zero treatment of total RNA. **h** In vitro assay monitoring Pol II elongation on an arrest sequence in the presence of TFIIS and increasing amounts of PHF3 (left) or in the presence of PHF3 alone (right). Pol II-EC was formed using an excess of a DNA–RNA bubble scaffold containing 5'-FAM-labeled RNA. The short elongation product seen in the 'no NTP' lane is due to residual ATP from the phosphorylation reaction. The experiments were repeated three times and the representative gels are shown. **i** A model of PHF3-mediated regulation of backtracking through competition with TFIIS. PHF3 represses transcription by competing with TFIIS and impeding Pol II rescue from backtracking, which may result in premature termination.

with TFIIS for binding to Pol II and thereby impedes the rescue of backtracked Pol II. Pol II backtracking is a widespread phenomenon and backtracked Pol II may be particularly susceptible to premature termination and fast turnover[17], which would explain why PHF3-regulated genes are low-expressed with low Pol II occupancy, but have open promoters (Supplementary Fig. 13). Our data suggest that PHF3 represses these genes by competing with TFIIS to prevent the rescue of backtracked Pol II and promote premature termination. Structural and biochemical analysis of TFIIS in complex with Pol II showed that TFIIS establishes multiple contacts with the Pol II elongation complex: domain II-III binds the Pol II funnel and pore while domain I binds the PAF1 complex[35,83]. Our biochemical experiments showed that functional competition between PHF3 and TFIIS requires PHF3 TLD, which occupies the same position on the Pol II complex as TFIIS domain II. Although PHF3 displaces TFIIS from the Pol II pore, TFIIS may remain associated with the Pol II elongation complex through its interaction with PAF1C. In the absence of PHF3, TFIIS binds more strongly to Pol II lobe-jaw domain and stimulates productive elongation (Fig. 7i). Ongoing structural work with the full-length PHF3 is expected to further clarify the competitive binding mechanism of PHF3 with respect to TFIIS.

How does PHF3 regulate mRNA stability? In the context of global effects of PHF3 loss on reduced elongation, the observed increase in mRNA stability may be a compensatory mechanism to maintain steady-state transcript levels, also referred to as transcript buffering[84–86]. Moreover, a recent study in yeast showed that Pol II elongation rate correlates negatively with mRNA stability[87]. Negative correlation between elongation rate and mRNA stability in PHF3 KO/ΔSPOC cells would suggest that PHF3 is directly or indirectly involved in reinforcing the negative feedback loop between transcription and mRNA stability. As a direct regulator, PHF3 may cooperate or compete with elongation and RNA processing factors that bind to the Pol II RPB4/7 stalk, which coordinates transcription elongation with mRNA decay[86]. In the context of a select number of derepressed genes, PHF3 may regulate mRNA stability by modulating the recruitment of RNA-binding proteins (RBPs) that were identified in the PHF3 interactome and/or that associate with the Pol II CTD and regulate RNA processing. RNA processing factors could require PHF3 for binding to the CTD. PHF3 interacts with alternative splicing factors such as SON, ZNF326, SAFB, RBMX, and RBM15

(Fig. 1a), but the analysis of RNA-seq data did not reveal any major changes in splicing due to PHF3 loss. PHF3 interaction with RBM15, which is part of the m6A writer complex, may affect m6A RNA levels and thereby modulate mRNA stability[88]. Furthermore, PHF3 interacts strongly with SPT6, which was shown to regulate mRNA stability through the CCR4-NOT complex in yeast[89]. CCR4-NOT promotes transcription elongation and deadenylates mRNAs as the first step in mRNA decay. Given that SPT6 facilitates CCR4-NOT recruitment[89], PHF3 might promote CCR4-NOT recruitment onto Pol II via SPT6 and thereby promote mRNA degradation.

Coupling between elongation and RNA processing through Pol II CTD becomes even more relevant considering recent findings that the CTD undergoes liquid-liquid phase separation (LLPS)[29,90]. LLPS of unphosphorylated CTD may facilitate Pol II clustering and transcriptional bursting during transcription initiation, whereas CTD phosphorylation dissolves initiation clusters and drives Pol II clustering with RNA processing factors[30]. Our findings that PHF3 condensates capture phPol II suggest that PHF3 may act as a bridge between pS2 CTD and RNA processing factors. While further work will elucidate how exactly PHF3 regulates mRNA stability, our results establish PHF3 as a mammalian synthegradase that coordinates transcription with mRNA decay[91,92].

Why do genes react differently to PHF3 loss? Neuronal genes were enriched among de-repressed genes with high levels of nascent and/or mature transcripts. These genes are low-expressed and Polycomb-repressed in WT cells but bear the marks of poised Pol II (H3K4me3, Pol II pS5). Poised Pol II is found in ESCs as well as differentiating and post-mitotic cells[51]. During neuronal differentiation, poised Pol II primes neuronal transcription factors for activation whilst keeping non-neuronal genes silenced[51]. PHF3 may prevent efficient TFIIS-mediated rescue of poised Pol II from backtracking and induce premature termination. Reactivation of these genes would thus be highly dependent on TFIIS, which may be the reason for their marked derepression in PHF3 KO cells where TFIIS would gain more access to Pol II.

PHF3 shows ubiquitous expression across tissues with the lowest expression in the brain (Supplementary Fig. 19a). Similar expression pattern and function was demonstrated for small CTD phosphatases (SCPs) specific for Pol II CTD pS5[93]. SCPs are recruited by the repressor element 1 (RE-1)–silencing transcription factor/neuron-restrictive silencer factor (REST/NRSF)

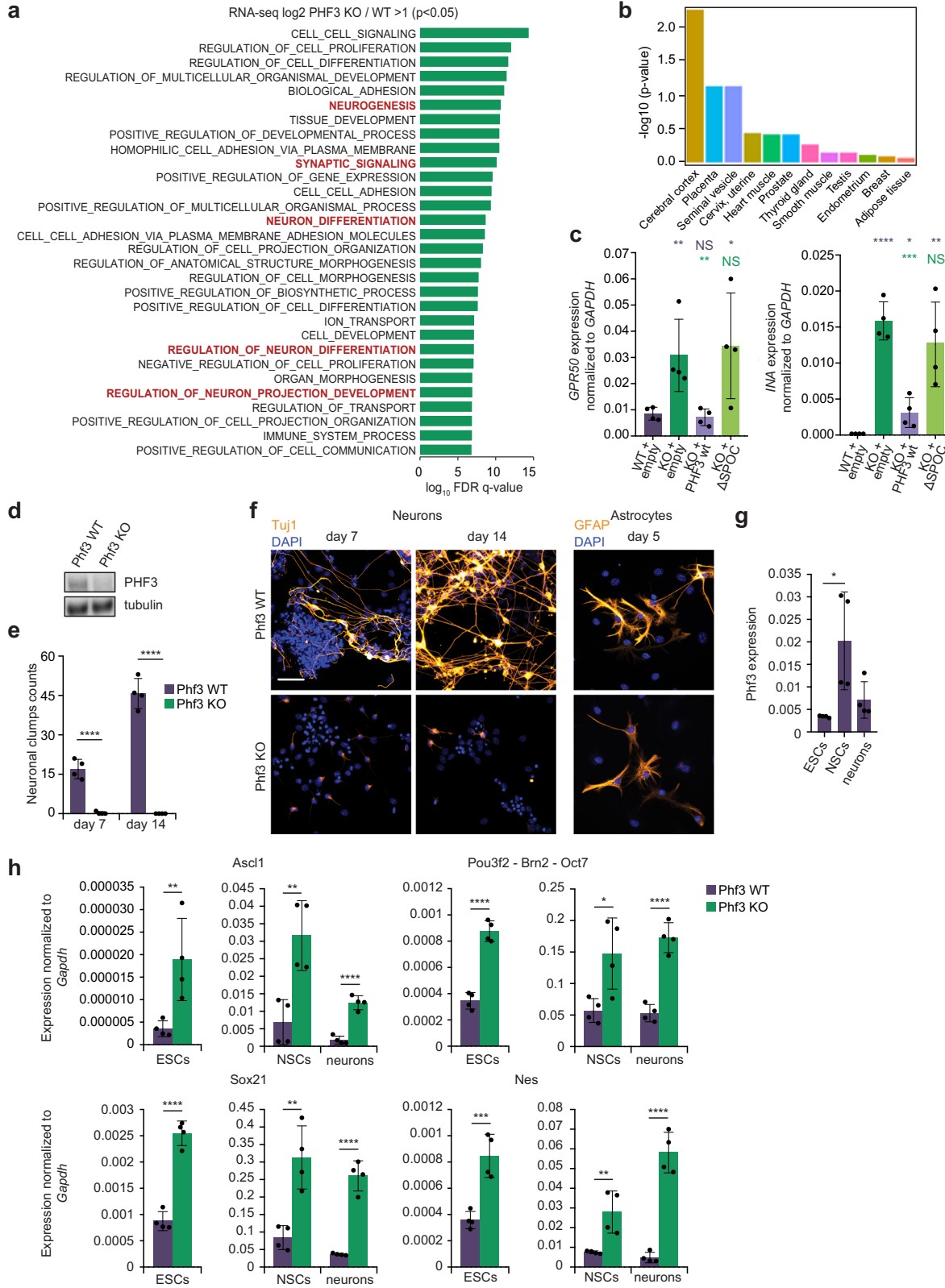

complex to neuronal genes and facilitate neuronal gene repression[93]. However, we did not detect any correlation between PHF3-mediated changes in gene expression and publicly available REST binding data for HEK293 cells (Supplementary Fig. 20). Instead, PHF3-repressed genes show enrichment of the Polycomb H3K27me3 mark. Conditional deletion of the H3K27me3 methylatransferase Ezh2 in the developing mouse midbrain leads

to derepression of several forebrain specification genes[94]. Taken together, SCPs and PHF3 may engage in two different pathways, REST and Polycomb, which independently protect from premature or ectopic expression of neuronal genes[95,96].

During neuronal differentiation of mESCs, Phf3 fine-tunes expression of neuronal genes to ensure their timely and adequate expression levels during differentiation. Phf3 KO mESCs fail to

**Fig. 8 PHF3 regulates neuronal gene expression and neuronal differentiation of mouse embryonic stem cells (mESCs). a** GO analysis of genes upregulated in PHF3 KO HEK293T cells according to RNA-seq shows enrichment of genes involved in neurogenesis. GSEA Biological processes tool was used. **b** TissueEnrich analysis shows the enrichment of cerebral cortex among transcriptionally upregulated genes in PHF3 KO. TissueEnrich analyses the enrichment of a particular gene set in the tissue specific expression profiles provided by the GTEx transcriptional compendium (https://www.gtexportal.org/home/faq#citePortal). The y-axis shows the -log10 of Benjamini–Hochberg corrected *p*-value. **c** RT-qPCR analysis of *INA* and *GPR50* mRNA levels in PHF3 WT and PHF3 KO HEK293T cells with stable integration of mCherry empty vector, and KO-complemented cell lines stably expressing mCherry-PHF3 wild-type or ΔSPOC. Four biologically independent experiments were performed. Data are presented as mean values ± standard deviation. The bars represent average expression from different clones as biological replicates. A t-test was performed by comparing expression levels with WT (violet asterisk) and KO (green asterisk). **d** CRISPR/Cas9 Phf3 knock-out in mESCs shows complete loss of protein by Western blotting. The experiment was performed once. **e** Quantification of beta III tubulin (TuJ1)-positive neuronal clump formation in Phf3 WT and KO cells after 7 or 14 days of neuronal differentiation. Neuronal clumps represent agglomerates of cells connected with Tuj1-positive cell projections. Four biologically independent experiments were performed. Data are presented as mean values ± standard deviation. **f** Representative immunofluorescence images of TuJ1-stained neurons and glial fibrillary acidic protein (GFAP)-stained astrocytes. Scale bar = 40 μm. The experiment was performed four times. **g** Phf3 expression levels in embryonic stem cells (ESCs), neural stem cells (NSCs) and neurons determined by RT-qPCR. Four biologically independent experiments were performed. Data are presented as mean values ± standard deviation. **h** Comparison of expression levels of different neuronal markers between Phf3 WT and KO ESCs, NSCs, and neurons by RT-qPCR. Four biologically independent experiments were performed. Data are presented as mean values ± standard deviation. One-tailed, two-sample equal variance t-test was used to determine *p*-values in **c**, **e**, **g**, **h**. *P*-values are indicated in Supplementary Data 7.

---

undergo neuronal differentiation, implying that PHF3 is required for neuronal development. Indeed, a Phf3 KO mouse generated by the International Mouse Phenotyping Consortium (IMPC) exhibits neuronal dysfunction in the form of impaired auditory brainstem response and impaired startle reflex (www.mousephenotype.org). Intriguingly, Phf3 loss had no effect on astrocyte differentiation. Based on the Allen brain cell atlas as the most comprehensive database of expression profiles from human and mouse brain cell types[97,98] we observed that PHF3 is not expressed in non-neuronal cell types such as astrocytes, oligodendrocytes, endothelial cells and microglia (Supplementary Fig. 19b, c). This may explain why Phf3 loss had no effect on astrocytes.

PHF3 function in the regulation of transcription and mRNA stability may be important beyond development. PHF3 was identified as an autism-risk gene[99] due to deletion mutations in the linker region between its TLD and SPOC domain resulting in a frameshift and premature termination, producing a SPOC-less protein. This illustrates how the SPOC domain is essential for the function of PHF3, in accordance with our findings whereby PHF3 SPOC deletion phenocopies PHF3 knock-out. Moreover, PHF3 expression levels were found to be significantly downregulated in glioblastoma[61], while ASCL1, POU3F2, and SOX2, which were all derepressed in Phf3 KO cells, have been implicated in the maintenance and tumorigenicity of glioblastoma[100,101]. Our data suggest that PHF3 downregulation may drive glioblastoma via derepression of transcription factors that regulate neuronal differentiation.

## Methods

**Cell lines and cell culture**. HEK293T cells were grown in Dulbecco's Modified Eagle's Medium (DMEM 4.5 g/L glucose) (Sigma) supplemented with 10% fetal bovine serum (Sigma), 1% L-glutamine (Sigma), 1% penicillin-streptomycin (Sigma) under 5% $CO_2$ at 37 °C. mESCs were cultured on 0.2% gelatin coated plates in ES-DMEM medium supplemented with LIF and 2i[102]. To generate CRISPR/Cas9 PHF3 KO, gRNA targeting exon 3 was cloned between BbsI sites under the U6 promoter in the plasmid encoding Cas9-EGFP (pX458)[103]. gRNA sequence for human PHF3 was 5′-TGATACTAGTACTTTTGGAC-3′ and for mouse Phf3 5′-ATTCGGGTTCCTCGGCCGTC-3′ (Supplementary Table 4). After 48 h of transfection with polyethylenimine (PEI; Polysciences), GFP-positive HEK293T cells were FACS-sorted and allowed to recover in culture for 4–7 days. Cells were subsequently FACS-sorted and GFP-negative cells were seeded 1 cell/well in 96-wells plates. After 14 to 20 days, surviving clones were expanded in culture, genomic DNA was isolated and PCR-amplified Cas9 target region was sequenced. To generate CRISPR/Cas9 Phf3 KO in mESCs, Rex1GFPd2::Cas9 (RC9) ES cells were used that carry a destabilized GFP-reporter for Rex1-expression and a stably expressed Cas9 transgene integrated into the Rosa26 locus. ES cells were co-transfected with 0.5 μg gRNA-expressing plasmid and 0.1 μg of dsRed expressing plasmid. After 2 days dsRed-positive cells were FACS-sorted and plated at clonal density into 60 mm TC-dishes. After 7 days colonies were picked and expanded in 96-well plates. To identify KO clones, genomic DNA was isolated and PCR-

amplified Cas9 target region was sequenced. To generate CRISPR/Cas9 endogenously GFP-tagged PHF3, two gRNAs targeting 3′ PHF3 terminus were designed (5′-CAGTGTGGTCCCTATCTTTG-3′ and 5′-TAAAATTTGCAGGCTGCTTC-3′) and cloned into the plasmid pX335 encoding Cas9 nickase[104]. Plasmid-borne repair template consisted of EGFP-P2A-puromycin flanked by 1.5 kb sequences homologous to the targeted genomic region. HEK293T cells were transfected with 2 μg of each of the plasmids encoding Cas9 nickase and one of the two gRNAs and 4 μg of the repair template. Two weeks after transfection, GFP positive cells were sorted by FACS. Two days after the sorting 0.5 μg/mL puromycin was added to the culture medium. After 1–2 weeks, surviving colonies were picked and expanded, genomic DNA was extracted and positive clones were identified by PCR. To generate CRISPR/Cas9 PHF3 ΔSPOC, one gRNA target site on either side of the SPOC domain was selected in such a way that the PAMs are within the deleted fragment (5′-TGGCTCGATTGAACTTCATC-3′ and 5′-GGTCCATCAAAAGG-CACAAG-3′). A 150 bp ssDNA repair template was designed which introduces an XhoI restriction site at the junction of the two breaks without shifting the reading frame or altering the amino acid sequence (5′-TTAGGCTTTTTAATGTCATTT TCTAGTCCAGAGATGCCTGGAACTGTTGAAGTTGAGTCTACCTTTCTGG CTCGAGTGCCTTTTGATGGACCTGGTAGGTATACGTTTTAATAATAGG ATAAGAATGAAATAACATGGGAGGTGGGACCA-3′). S-phase synchronized HEK293T cells were electroporated with 10 μg of purified Cas9, 12 μg of each in vitro transcribed gRNA and 4 μM repair template. Genomic DNA from the clones was PCR-amplified to check for the deletion of 4.5 kb and digested with XhoI to ensure that the repair template was used for homologous recombination-mediated repair. To generate stable cell lines expressing mCherry-PHF3 constructs, HEK293T PHF3 KO cells were transfected with 2 μg of plasmid and PEI in 6-well plates. After 48 h, half of the cells were transferred to 10 cm dishes and grown in medium supplemented with 0.25 μg/mL of puromycin. After 2–3 weeks, surviving colonies were picked using glass cylinders and monoclonal populations were expanded in culture. Positive clones were validated by Western blot.

**Constructs**. Human PHF3 was amplified from HEK293T cDNA and cloned into CMV10 N3XFLAG (Sigma) between NotI and XbaI. PHF3 truncation constructs were generated by tripartite ligation of BsaI-flanking fragments according to the Golden Gate cloning principle[105]. PHF3 and ΔSPOC constructs used for complementation were cloned into mCherry IRES puromycin vector (Clontech) between AgeI and NotI. Site-directed mutagenesis was performed according to the FastCloning protocol[106]. For bacterial expression, PHF3 SPOC domain (1199–1356aa) and PHF3 TLD (924–1046aa) were cloned into pET M11 between NcoI and XhoI for N-terminal $His_6$ fusion. PHF3 SPOC was additionally cloned into pET Duet 6xHis-TEV-mCherry between SacI and NotI. For insect cell expression, PHF3 was cloned into pFB32 for N-terminal $His_6$ fusion and C-terminal Strep fusion. TFIIS mutant (TFIIS^M) was generated by site-directed mutagenesis to introduce D282A E283A mutations in pOPINB TFIIS for bacterial expression. pK10R7Sumo-3C TFIIF was generated by amplifying RAP74 and RAP30 from HEK293T cDNA; RAP74 was cloned between BamHI and NotI for N-terminal SUMO-$His_{10}$ fusion, RAP30 was cloned between NdeI and KpnI. Codon optimized mEGFP-CTD was kindly provided by Marek Sebesta.

**Protein purification**. SPOC and TLD were expressed in *E. coli* Rosetta2 (DE3) cells (Novagen) and purified by affinity chromatography using HisTrap HP column (GE Healthcare) equilibrated in 25 mM Tris-Cl pH 7.4, 500 mM NaCl, 20 mM imidazole, followed by TEV cleavage of the $His_6$ tag and size exclusion chromatography using Sephacryl S-200 (GE Healthcare) equilibrated in 25 mM Tris-Cl pH 7.4, 25 mM NaCl and 1 mM DTT. TFIIS or TFIIS^M were expressed in *E. coli* Rosetta2

(DE3) cells (Novagen) and purified by affinity chromatography using HisTrap HP column (GE Healthcare) equilibrated in 25 mM Tris-Cl pH 7.4, 500 mM NaCl, 20 mM imidazole, and by size exclusion chromatography using Sephacryl S-200 (GE Healthcare) equilibrated in 5 mM Hepes pH 7.25, 100 mM NaCl, 10 μM ZnCl$_2$ and 10 mM DTT. TFIIF (RAP74/RAP30) was expressed in *E. coli* Rosetta2 (DE3) cells (Novagen) and purified by affinity chromatography using HisTrap HP column (Ge Healthcare) equilibrated in 25 mM Tris-Cl pH 7.4, 500 mM NaCl, 20 mM imidazole, followed by 3 C cleavage of the SUMO-His$_{10}$ tag, cation exchange chromatography using HiTrap SP column (GE Healthcare) equilibrated in 50 mM Hepes pH 7, 150 mM KCl, 10% glycerol and 2 mM DTT, and by size exclusion chromatography using Sephacryl S-200 (GE Healthcare) equilibrated in 20 mM Hepes pH 7.5, 100 mM NaCl, 10% glycerol and 2 mM DTT. mEGFP-CTD was expressed in *E. coli* Rosetta2 (DE3) cells and purified by affinity chromatography using Ni-NTA beads equilibrated in 20 mM Tris-Cl pH 7.4, 500 mM NaCl, 20 mM imidazole, 10% glycerol in the presence of protease inhibitors, followed by size exclusion chromatography using Superdex 75 (GE Healthcare) equilibrated in 20 mM Hepes pH 7.4, 250 mM NaCl, 10% glycerol and 1 mM TCEP. PHF3 was expressed from the EMBacY bacmid in Sf9 cells with an N-terminal His-tag and a C-terminal Strep-tag. PHF3 was purified by affinity chromatography using HisTrap FF column (GE Healthcare) equilibrated in 50 mM Tris-Cl pH 8, 300 mM NaCl, 0.5 mM TCEP, 20 mM imidazole, followed by anion exchange chromatography using HiTrap ANX (high sub) FF column (GE Healthcare) equilibrated in 50 mM Tris-Cl pH 8, 100 mM NaCl, 0.5 mM TCEP, 10% glycerol, and size exclusion chromatography using Superose 6 (GE Healthcare) equilibrated in 50 mM Tris-Cl pH 8.0, 300 mM NaCl, 0.5 mM TCEP. Pol II was purified from pig thymus, as previously described[107]. Pig thymus was sourced from animals approved for food consumption through an officially approved facility in Sieghartskirchen, Lower Austria. PHF3 was labeled with Alexa Fluor 488 or 594, while Pol II was labeled with Alexa Fluor 647 Conjugation Kit (Fast)—Lightning-Link (Abcam) according to the manufacturer's instructions and purified over a Superdex 200 Increase 3.2/200 column (GE Healthcare).

**Immunoprecipitation.** For immunoprecipitation of exogenously expressed FLAG-PHF3 constructs, a 10 cm dish of transfected HEK293T cells was used. Cells were harvested 48 h after transfection and lysed in lysis buffer (50 mM Tris-Cl pH 8, 150 mM NaCl, 1% Triton, 1× protease inhibitors, 2 mM Na$_3$VO$_4$, 1 mM PMSF, 2 mM NaF, 50 units/mL benzonase and 1 mM DTT) for 1 h at 4 °C. 10% of the cleared lysate was kept as input and the rest was incubated for 2 h on a rotating wheel at 4 °C with anti-FLAG M2 magnetic beads (Sigma). For Pol II IP, Protein G Dynabeads (Invitrogen) were washed twice with TBS and incubated with 5 μg of mouse anti-FLAG M2 (Sigma), rabbit anti-pS2 Pol II (Abcam ab5095), mouse anti-pS5 Pol II 4H8 (Abcam ab5408), rat anti-pS7 Pol II clone 4E12 (Millipore) or mouse anti-Pol II clone F-12 (Santa Cruz) antibodies for 1 h on a rotating wheel at room temperature. Beads were washed twice with TBS and cleared lysates were added for immunoprecipitation on a rotating wheel at 4 °C ON. For immuno-precipitation of endogenously tagged PHF3-GFP, two 15 cm dishes were used for each cell line. Cells were harvested and lysed in lysis buffer (as above but without DTT). The lysates were incubated on a rotating wheel at 4 °C ON with rabbit anti-GFP (Abcam ab290) antibody. The samples were added to protein G Dynabeads (Invitrogen) and incubated on a rotating wheel at 4 °C for 6 h. Beads were sub-sequently washed three times with TBS and immunoprecipitated proteins were eluted twice with 0.1 M glycine pH 2 and neutralized with Tris-Cl pH 9.2. For Western blot, 2% of the input and 20% of the eluate were loaded for each sample. Western blots were analyzed using Image Lab 6.0.1 (Biorad). For mass spectrometry analysis of FLAG-PHF3 or Pol II interactome, immunoprecipitations were performed as described above and samples were processed for on beads digestion.

**Mass spectrometry.** Beads were eluted three times with 20 μL 100 mM glycine and the combined eluates were adjusted to pH 8 using 1 M Tris-HCl pH 8. Disulfide bonds were reduced with 10 mM DTT for 30 min before adding 25 mM iodoacetamide and incubating for another 30 min at room temperature in the dark. The remaining iodoacetamide was quenched by adding 5 mM DTT and the proteins were digested with 300 ng trypsin (Trypsin Gold, Promega) overnight at 37 °C. The digest was stopped by addition of 1% trifluoroacetic acid (TFA), and the peptides were desalted using C18 Stagetips. NanoLC-MS analysis was performed using the UltiMate 3000 HPLC RSLC nano system (Thermo Scientific) coupled to a Q Exactive mass spectrometer (Thermo Scientific), equipped with a Proxeon nanospray source (Thermo Scientific). For FLAG-PHF3 IP samples, peptides were loaded onto a trap column (PepMap C18, 5 mm × 300 μm ID, 5 μm particles, 100 Å pore size; Thermo Scientific) followed by the analytical column (PepMap C18, 500 mm × 75 μm ID, 3 μm, 100 Å; Thermo Scientific). The elution gradient started with the mobile phases: 98% A (water/formic acid, 99.9/0.1, v/v) and 2% B (water/acetonitrile/formic acid, 19.92/80/0.08, v/v/v), increased to 35% B over the next 120 min followed by a 5-min gradient to 90% B, stayed there for 5 min and decreased in 5 min back to the gradient 98% A and 2% B for equilibration at 30 °C. The Q Exactive mass spectrometer was operated in data-dependent mode, using a full scan followed by MS/MS scans of the 12 most abundant ions. For peptide identification, the.RAW-files were loaded into Proteome Discoverer (version 1.4.0.288, Thermo Scientific). The resultant MS/MS spectra were searched using Mascot 2.2.07 (Matrix Science) against the Swissprot protein sequence database,

using the taxonomy human. The peptide mass tolerance was set to ±5 ppm and the fragment mass tolerance to ±0.03 Da. The maximal number of missed cleavages was set to 2. The result was filtered to 1% FDR using Percolator algorithm integrated in Proteome Discoverer[108]. For Pol II IP samples, a pre-column for sample loading (Acclaim PepMap C18, 2 cm × 0.1 mm, 5 μm, Thermo Scientific), and a C18 analytical column (Acclaim PepMap C18, 50 cm × 0.75 mm, 2 μm, Thermo Scientific) were used, applying a segmented linear gradient from 2 to 35% and finally 80% solvent B (80% acetonitrile, 0.1% formic acid; solvent A 0.1% formic acid) at a flow rate of 230 nL/min over 120 min. Eluting peptides were analyzed on a Q Exactive HF-X Orbitrap mass spectrometer (Thermo Scientific), which was coupled to the column with a customized nano-spray EASY-Spray ion-source (Thermo Scientific) using coated emitter tips (New Objective). The mass spectrometer was operated in data-dependent acquisition mode (DDA), survey scans were obtained in a mass range of 375–1500 m/z with lock mass activated, at a resolution of 120k at 200 m/z and an AGC target value of 3E6. The 8 most intense ions were selected with an isolation width of 1.6 m/z, fragmented in the HCD cell at 27% collision energy and the spectra recorded for max. 250 ms at a target value of 1E5 and a resolution of 30k. Peptides with a charge of +1 or > +6 were excluded from fragmentation, the peptide match feature was set to preferred, the exclude isotope feature was enabled, and selected precursors were dynamically excluded from repeated sampling for 20 s. Raw data were processed using the MaxQuant software package (version 1.6.0.16;[109]) and the Uniprot human reference proteome (July 2018, www.uniprot.org) as well as a database of most common contaminants. The search was performed with full trypsin specificity and a maximum of three missed cleavages at a protein and peptide spectrum match false discovery rate of 1%. Carbamidomethylation of cysteine residues were set as fixed, oxidation of methionine, phosphorylation of serine, threonine, and tyrosine, and N-terminal acetylation as variable modifications.

**Analysis of mass spectrometry data.** For the analysis of the PHF3 interactome (FLAG-PHF3 IP), SAINT-MS1 was used as a statistical tool to determine the probability of protein–protein interactions[110]. Prior to analysis with SAINT-MS1[110] the label-free quantification data were cleaned by removing bait and common laboratory contaminants[111]. The control (empty vector) was used simultaneously to estimate the parameters of the false interaction probability distributions. SAINT-MS1 was run for each method and fraction separately with 5000 and 10000 burn-in and sampling iterations, respectively. Protein areas were normalized to obtain a median protein ratio of one between samples. Fold changes were calculated based on these normalized protein areas. For the analysis of differential Pol II interactome between PHF3 WT and KO cells, label-free quantification the "match between runs" feature and the LFQ function were activated in the MaxQuant software package[109]. Downstream data analysis was performed using the LFQ values in Perseus (version 1.6.2.3;[109]). Mean LFQ intensities of biological replicate samples were calculated and proteins were filtered for at least two quantified values being present in the three biological replicates. Missing values were replaced with values randomly selected from a normal distribution (with a width of 0.3 and a median downshift of 1.8 standard deviations of the sample population). To determine differentially enriched proteins we used the LIMMA package in R (version 3.5.1.) and applied the Benjamini-Hochberg correction for multiple testing to generate adjusted p-values.

**Fluorescence anisotropy (FA).** All measurements were conducted on a FluoroMax-4 spectrofluorometer (Horiba Jobin-Yvon). The instrument was equipped with a thermostated cell holder with a Neslab RTE7 water bath (Thermo Scientific). The system was operated by FluorEssence software (version 2.5.3.0 and V3.5, Horiba Jobin-Yvon). All measurements were performed at 20 °C in 25 mM Tris-Cl pH 8, 25 mM NaCl, 1 mM DTT with the exception of measurements in Supplementary Fig. 2f where 25 mM and 100 mM NaCl buffer conditions were compared. CTD peptides were labeled N-terminally with 5,6-carboxyfluorescein (FAM λex = 467 nm, λem = 517 nm; Clonestar) (Supplementary Table 1). 10 nM CTD peptide (in a volume of 1.4 ml) was titrated with increasing amounts of SPOC protein. Each data point is an average of three measurements. The binding iso-therms were generated by non-liner regression analyses with the software package GraphPad Prism 7 (GraphPad Software, La Jolla).

**Fluorescence correlation spectroscopy (FCS).** In total 0.2 mg of lyophilized 2xpS2 and 2xpS2pS5 peptides (Clonestar) were dissolved in 30 μl DMSO (Sigma-Aldrich). One equivalent of Atto-488 NHS ester (Atto-tec GmbH) was added to three equivalents of peptide in DMSO followed by five equivalents of DIPEA (Sigma-Aldrich) and incubated for 16 h at room temperature protected from light. The reaction mixture was diluted to 50% (v/v) DMSO with water and purified by reverse-phase HPLC on a C18 column (Agilent Technologies) using a 10–70% gradient of acetonitrile + 0.1% TFA to water + 0.1% TFA over 16 min, followed by flushing at 90% acetonitrile + 0.1% TFA for 2 min. The desired fractions were lyophilized and stored at −20 °C. A ConfoCor 2 spectrofluorimeter (Carl Zeiss-Evotec) equipped with an air-cooled Argon-laser (LASOS Lasertech GmbH; intensity 70 μW) and a water immersion objective (C-Apochromat 63×/1.2 W Corr) was used for monitoring changes in diffusion behavior due to binding of 15 nM labeled 2xpS2 or 2xpS2pS5 peptide to dilution series of 40–0.2 μM of SPOC.

All FCS measurements were performed in a 1536-well glassplate (Greiner Bio-One) and a sample volume of 5 μL at 20 °C. The diameter of the pinhole was set to 35 μm and the confocal volume was calibrated using 4 nM of Atto 488 dye ($D_{trans} = 4.0 \times 10^{-10}$ m$^2$ s$^{-1}$). Intensity fluctuations were recorded by an avalanche photodiode (SPCM-CD 3017) in photon counting mode over a time period of 15 s and repeated 8 times for each sample. The normalized autocorrelation function $G(\tau)$ describes the observed fluctuations of the fluorescence intensity $\delta F(t)$ from the mean intensity at any time compared to fluctuations at time $t + \tau$. It is given by

$$G(\tau) = 1 + \frac{\langle \delta F(t) \delta F(t + \tau) \rangle}{\langle F \rangle^2}$$

where the angular brackets represent the ensemble average, $\langle F \rangle$ denotes the mean intensity, and $\tau$ is known as the delay or correlation time interval over which the fluctuations are compared. For a single diffusing species of Brownian motion in a 3D Gaussian confocal volume element with half axes $\omega_{xy}$ and $\omega_z$, the auto-correlation function $G(\tau)$ is defined by

$$G(\tau) = 1 + \frac{1}{N}\left(1 + \frac{\tau}{\tau_D}\right)^{-1}\left(1 + \left(\frac{\omega_{xy}}{\omega_z}\right)^2 \frac{\tau}{\tau_D}\right)^{-\frac{1}{2}}$$

where $N$ is the number of particles, $\tau_D = \frac{\omega_{xy}^2}{4D}$, $\tau_D$ being the molecular diffusion time of the excited fluorophores moving in a three-dimensional confocal volume through an axial ($z$) to radial ($xy$) dimension, and $D$ the diffusion coefficient [cm$^2$/s]. Evaluation of the autocorrelated curves was performed with the FCS ACCESS Fit (Carl Zeiss-Evotec) software package using a Marquardt nonlinear least-squares algorithm for a one-component fitting model[112]. The average hydrodynamic radius $R_h$ of the protein was calculated from the obtained translational diffusion coefficient $D_{trans}$ using the Stokes-Einstein relation

$$R_h = \frac{k_B T}{6\pi\eta D_{trans}}$$

where $k_B$ is the Boltzmann constant ($1.38 \times 10^{-23}$ J/K), $T$ is the temperature (293 K), and $\eta$ is the viscosity of the solvent (0.001 kg m$^{-1}$ s$^{-1}$). The molecular weight of the protein was estimated by

$$M_W = \frac{4}{3}R_h^3 \pi \rho N_A$$

where $N_A$ is Avogadro's number = $6.023 \times 10^{23}$ mol$^{-1}$, and $\rho$ is the mean density of the molecule. By titrating 40–0.2 μM SPOC to 15 nM of Atto-488 labeled peptides 2xpS2 or 2xpS2pS5, the diffusion times significantly increased from 74.4 ± 2.7 μsec (2xpS2) or 75.0 ± 0.9 μsec (2xpS2pS5) for peptide alone to 133.0 ± 7.8 μsec (2xpS2-SPOC) or 127.0 ± 5.0 μsec (2xpS2pS5-SPOC) for SPOC-bound peptides (Supplementary Fig. 3). The calculated corresponding molar masses of 1.9 ± 0.2 kDa (2xpS2), 2.0 ± 0.1 kDa (2xpS2pS5), 17.9 ± 3.6 kDa (SPOC), 20.3 ± 3.8 kDa (2xpS2-SPOC) and 17.7 ± 2.2 (2xpS2pS5-SPOC) clearly showed a 1:1 binding stoichiometry (Supplementary Fig. 3c).

**Mass photometry.** Mass photometry measurements were performed on a Refeyn OneMP mass photometer using AcquireMP software (v2.3.0). Marienfeld high precision glass coverslips (24 × 50 mm, No. 1.5H) were cleaned by sequential sonication in Milli-Q H20, isopropanol (HPLC grade), and Milli-Q H2O (5 min each), followed by drying with a clean nitrogen stream. Clean coverslips were equipped with self-adhesive silicone culture wells (Grace Bio-Labs reusable CultureWellsTM gaskets). In total 18 μl buffer (50 mM Tris pH 8, 300 mM NaCl, 0.5 mM TCEP) were pipetted into a culture well and the focal position was identified, then PHF3 was added to a final concentration of 20 nM and movies of 60 s duration were recorded using default settings and medium field of view (imaged area: 10.8 μm × 6.8 μm). Mass photometry data were processed and analyzed using DiscoverMP software (v2.4.0). Contrast-to-mass calibration was performed using Invitrogen NativeMarkTM Unstained protein standard containing proteins of the masses 1048 kDa, 480 kDa and 146 kDa.

**X-ray crystallography.** Due to the low sequence identity with published SPOC domain structures of human SHARP and Arabidopsis FPA, PHF3 SPOC structure was solved using the single-wavelength anomalous diffraction (SAD) method. Initial crystals of SPOC at 5 mg/mL were obtained using the sitting-drop vapor diffusion technique and a nanodrop-dispensing robot (Phoenix RE; Rigaku Europe). Crystallization conditions were optimized using microseed matrix screening approach (MMS)[113]. The best diffracting crystals were grown in conditions B3 from ShotGun HT screen (SG1 HT96 Molecular Dimensions, Suffolk, UK) containing 0.2 M MgCl$_2$, 0.1 M Bis-Tris pH 5.5, 25% PEG 3350 at 22 °C. For co-crystal structures, a 3-fold molar excess of 2xpS2 (Clonestar), 2xpS2pS5 (Clonestar) and 2xpS2pS7 CTD (Eurogentec) peptides was incubated with 5 mg/mL of SPOC. Co-crystals were grown using the sitting-drop vapor diffusion technique. The best diffracting crystals of SPOC:2xpS2 were obtained using MMS approach in Morpheus screen E9 conditions (Morpheus HT, Molecular Dimensions) containing 0.12 M ethylene glycol mixture, 0.1 M Tris-Bicine buffer pH 8.5, 20% glycerol and 10% PEG 4000; crystals of SPOC:pS2pS7 were obtained in JCSG C4 conditions (JCSG HT96, Molecular Dimensions) containing 0.1 M Hepes pH 7.0, 10% PEG 6000; crystals of SPOC: 2xpS2pS5 were obtained in Morpheus A3 conditions

containing 0.03 M MgCl$_2$, 0.03 M CaCl$_2$, 0.1 M imidazole-MES pH 6.5, 20% glycerol and 10% PEG 4000. The crystals were flash cooled in liquid nitrogen prior to data collection. The selenomethionine data set was collected at the beamline ID29 (ESRF, Grenoble) at 100 K at the peak of selenium using a wavelength of 0.979 Å. The data sets of SPOC-CTD peptide complexes were collected at the MASSIF beamline ID30a1 (ESRF, Grenoble) at 100 K using a wavelength of 0.966 Å. The data set of SPOC:2xpS2pS5 was collected at the beamline ID29 (ESRF, Grenoble) using a wavelength 1.07 Å. The data frames were processed using the XDS package[114], and converted to mtz format with the program AIMLESS[115]. The apo-SPOC structure was solved using single anomalous diffraction with the CRANK 2 software suite[116]. The structures of SPOC in complex with 2xpS2, 2xpS2pS5, and 2xpS2pS7 CTD peptides were solved using the molecular replacement program PHASER[117] with atomic coordinates of apo-SPOC as a search model. The structures were then refined with REFMAC[115,118] and Phenix Refine[119] and rebuilt using Coot[120]. The structures were validated and corrected using PDB_REDO server[121]. The figures were produced using the PyMol software. Coordinates were deposited in the protein data bank (accession codes: 6IC8 for PHF3 SPOC:2xpS2, 6IC9 for PHF3 SPOC:2xpS2pS7, 6Q2V for PHF3 SPOC, 6Q5Y for PHF3 SPOC:2xpS2pS5). Data collection and refinement statistics are reported in Supplementary Table 2. The crystal structure of 2xpS2pS5 CTD peptide bound to PHF3 SPOC showed a different binding mode compared to 2xpS2 and 2xpS2pS7 CTD peptides (Supplementary Fig. 2i–m). Two molecules of SPOC were bound to 2xpS2pS5 peptide, the conformation of the bound peptide was slightly different and three phosphorylated CTD residues (pS5 from the first heptapeptide and pS2pS5 from the second) were forming hydrogen bonds with SPOC residues (Supplementary Fig. 2i–m). SPOC residues K1267 and K1309 from the SPOC_C molecule and R1297 from SPOC_A formed hydrogen bonds with pS5$_a$; R1248 from SPOC_C with pS2$_b$; and R1248 and R1295 from SPOC_C and K1260 from SPOC_A hydrogen bonded with pS5$_b$) (Supplementary Fig. 2m). In sum, both positively charged patches from two SPOC molecules contributed interchangeably to the docking of the three phosphorylated serines (Supplementary Fig. 2m). While 2xpS2 structures indicate that the positive patches on the SPOC surface are geared to accommodate pS2 marks on adjacent repeats, 2xpS2pS5 structure reveals that the binding site can also adjust itself towards binding of phosphomarks within the same repeat. However, the FCS measurements showed a 1:1 binding stoichiometry for both 2xpS2 and 2xpS2pS5 peptides (Supplementary Fig. 3), confirming the data from the crystal structure of 2xpS2 and 2xpS2pS7. This indicates that 2:1 stoichiometry (SPOC:peptide) observed in the 2xpS2pS5 co-structure is most likely an artefact of crystal packing.

**In vitro condensate formation.** 4-well glass bottom slides (Ibidi) were coated with 1% PF127 (Sigma Aldrich) overnight at 4 °C and washed twice with 25 mM Tris-Cl pH 7.5, 50 mM NaCl, 1 mM DTT and 10% v/v PEG6000. Protein samples were loaded onto glass slides, mixed with the buffer to reach the final concentration of 20 mM Hepes pH 7.4, 150 mM NaCl, 1 mM TCEP, 10% dextran T500 (Pharmacosmos) and imaged within 15–45 min. PHF3 and Pol II were prepared by mixing Alexa labeled and unlabeled protein at 1:5 ratio. CTD and Pol II were phosphorylated with DYRK1A (generously provided by Matthias Geyer) in kinase buffer (50 mM Hepes pH 7.5, 34 mM KCl, 7 mM MgCl$_2$, 5 mM β-glycerophosphate, 2.5 mM DTT) with 1 mM ATP for 1 h at 30 °C. The reaction was desalted with PD MiniTrap G-25 (GE Healthcare) to remove MgCl$_2$ and ATP that could interfere with LLPS. Imaging was performed on Zeiss Axio Observer Z1 with a 60x oil immersion objective. Condensate size was analyzed using Fiji.

**In vitro fluorescence recovery after photobleaching (FRAP).** Condensates were prepared as described above and imaged on a Zeiss AxioObserverZ1 equipped with a Yokogawa CSU-X1-A1 Nipkow spinning disc unit (pinhole diameter 50 μm, spacing 253 μm) (Visitron) with an EM-CCD: back-illuminated evolve EM512 highspeed/high-resolution camera (Evolve™ EMCCD; Photometrics) using an EC Plan-NeoFluor 100x/1.30NA Oil objective lens. Equipment control and imaging was handled by Visiview software (version 5.0.0.11; Visitron). The 561 nm laser line was used at full laser intensity (200 mW) to photobleach a circular region within condensates to 40–50% of its initial intensity with 1 ms pixel dwell time. Fluorescence recovery was imaged every 300 ms over a period of 3 min using 561 nm laser line at 20% intensity. The fluorescence intensity of the bleached region was background-corrected and normalized to the fluorescence intensity prior to bleaching. Bleaching correction was based on measuring the fluorescence intensity in a similarly sized unbleached region within the condensate. The obtained recovery curves were fit to a single exponential recovery (one-phase association, $Y = Y0 + (\text{Plateau}-Y0)*(1-\exp(-K*x))$ where $Y$ represents the fluorescence intensity, $Y0$ the fluorescence intensity at timepoint 0, $K$ the rate constant and $x$ the time.) using GraphPad Prism 6.04.

**Immunofluorescence.** Cells were grown on glass coverslips, washed with PBS or with PEM buffer (100 mM Pipes, 5 mM EGTA, 2 mM MgCl$_2$, pH 6.8) for the neurons and astrocytes and fixed in 4% paraformaldehyde for 10 min. After rinsing with PBS, the cells were permeabilized with a 0.1% Triton X-100 solution in PBS for 8 min, rinsed again and blocked for 1 h RT in blocking buffer (0.1% Tween, 1% BSA in PBS). Coverslips were incubated with the primary antibodies for 1 h RT,

washed and subsequently incubated with secondary Alexa-conjugated antibodies for 1 h RT. After washing, coverslips were stained with DAPI and mounted on glass slides in ProLong Gold antifade reagent (Life technologies). Images were acquired using an LSM710 confocal microscope and processed with ImageJ software.

**EU incorporation assay.** HEK293T PHF3 WT, KO and ΔSPOC cells were grown for 24 h in 24-well plates for FACS analysis or on coverslips for immuno-fluorescence. Cells were then incubated with 0.5 μM EU (Molecular Probes) for 1 h. For immunofluorescence, cells were fixed in 2% PFA, washed in 3% BSA in PBS and permeabilized in 0.5% Triton X-100 in PBS. Click-iT® reaction was performed to couple Alexa Fluor 488 Azide (Molecular Probes) to the incorporated EU. Cells were subsequently stained with DAPI and coverslips were mounted on glass slides with ProLong Gold. For FACS analysis, cells were harvested by trypsinization, washed in PBS and fixed overnight in 75% methanol at −20 °C. Fixed cells were washed with PBS, blocked in 3% BSA and permeabilized in 0.25% Triton X-100 in PBS. Click-iT® reaction was performed to couple Alexa Fluor 488 Azide to the incorporated EU. Cells were subsequently washed twice in 3% BSA in PBS and finally resuspended in PBS. FACS measurements were performed on BD Fortessa machine using Diva software. A population of approximately $10^4$ cells was analyzed for each sample and cell counts in the gated P1 population were measured for three independent experiments using Flowing Software version 2.5.1. Cell counts for each fluorescence intensity were also exported in Microsoft Excel 2010 as frequency distributions of arbitrary fluorescence unit values. Average distributions of three independent experiments were plotted to generate the final FACS data histograms.

**High-resolution Airyscan imaging.** Cells were grown for at least 16 h on high precision glass coverslips (#1.5) coated with 0.5 μg/ml fibronectin in PBS for 2 h at RT. For EU incorporation assay cells were incubated with 0.5 μM EU (Molecular Probes) for 1 h, washed in PBS and fixed in 2% PFA for 15 min at RT. Fixed cells were washed twice in 3% BSA in PBS, permeabilized in 0.5% Triton X-100 in PBS for 10 min and washed twice in PBS. Click-iT® reaction was performed to couple Alexa Fluor 647 Azide (Molecular Probes) to the incorporated EU. For IF cover-slips were incubated with primary antibodies in 3% BSA, 0.1% Tween in PBS ON at 4 °C and subsequently incubated with Alexa Fluor secondary antibodies for 1 h at RT, counterstained with DAPI and mounted in ProLong Diamond antifade reagent (Life Technologies). Airyscan imaging was perfomed on an inverted Zeiss LSM 980 confocal microscope equipped with a 63x/1.4 Oil DIC objective and a 32 channel GaAsp Airyscan 2 detector unit controlled by Zen Black (software version 3.2). Sequential acquisitions of up to 4 channels were performed with 30 mW laser diodes (405 nm and 488 nm), a 25 mW DPSS laser (561 nm) and a 25 mW HeNe laser (639 nm) set to 1–3% excitation power. Images were captured with 2x Nquist settings (pixel size 40 nm, z interval 150 nm) with a pixel dwell time of ~ 6.6 μs and processed for superresolution with Airyscan filter 6. The resulting 3D high-resolution images were analyzed with the Zeiss co-localization plugin of Zen 3.2. Individual nuclei identified by DAPI were marked as regions of interest and thresholds were determined by Costes regression and randomization. Fluorescence intensity in co-localized clusters was quantified with Fiji 3D objects counter (ImageJ 1.53c). Colocalization coefficients for each cell were averaged across all slices of the z-stack and plotted for each cell individually. Box and whisker plots were generated from ~ $N = 20$ cells of at least two biological replicates.

**PRO-seq.** The protocol was adapted from Kwak et al, 2013. To isolate nuclei, cells were resuspended in cold buffer A (10 mM Tris-Cl pH 8, 300 mM sucrose, 3 mM $CaCl_2$, 2 mM MgAc₂, 0.1% TritonX-100, 0.5 mM DTT), incubated on ice for 5 min and transferred to a dounce homogenizer. After douncing 25 times with the loose pestle, cells were centrifuged at 700 g for 5 min. The pellet was resuspended again in buffer A, centrifuged and resuspended in buffer D (10 mM Tris-Cl pH 8, 25% glycerol, 5 mM MgAc₂, 0.1 mM EDTA, 5 mM DTT), flash frozen with liquid nitrogen and stored at −80 °C. For each run-on, $10^7$ HEK293 nuclei were mixed with $10^6$ of Drosophila S2 nuclei (10% for spike-in normalization) in 100 μL buffer D and incubated at 30 °C for 3 min with 0.025 mM biotin-11-NTPs and run-on master mix (5 mM Tris-Cl pH 8, 2.5 mM MgCl₂, 0.5 mM DTT, 150 mM KCl, 0.2 units/μL SUPERase In, 0.5% Sarkosyl). Nascent RNA was isolated using TRIzol LS reagent according to the manufacturer's instructions, denatured at 65 °C for 40 s, hydrolyzed using 0.2 M NaOH on ice for 20 min and neutralized with 1 volume of 1 M Tris-Cl pH 6.8. Buffer was exchanged with DEPC water using BioRad P-30 columns. Fragmented nascent RNA was subsequently enriched using Streptavidin M280 beads by rotating the samples for 20 min in binding buffer (10 mM Tris-Cl 7.4, 300 mM NaCl, 0.1% TritonX-100). Beads were subsequently washed twice with high-salt wash buffer (50 mM Tris-Cl pH 7.4, 2 M NaCl, 0.5% TritonX-100), twice with binding buffer, and once with low-salt wash buffer (5 mM Tris-Cl pH 7.4, 0.1% TritonX-100). RNA was isolated from the beads using TRIzol reagent in two consecutive rounds and pooled together for ethanol precipitation. The RNA pellet was redissolved in DEPC H₂O with 10 pmol of reverse 3′ RNA adaptor starting with a 5' random octamer sequence (5′- 5Phospho rNrNrNrNrNrNrNrNrGrAr-UrCrGrUrCrGrGrArCrUrGrUrArGrArArCrUrCrUrGrArArC-/inverted dT/ −3′) and subjected to ligation using T4 RNA ligase I (NEB) at 16 °C ON. RNA was isolated using Streptavidin M280 beads as previously described and 5′ ends were repaired using Cap-Clip Acid Pyrophosphatase (Cellscript) and Polynucleotide

Kinase (PNK, NEB) according to manufacturers' instructions. RNA was purified again with TRIzol and ethanol precipitation as before and subjected to 5′ RNA adaptor ligation as for the 3′ adaptor ligation (the 5′ adaptor contained a 3′ random tetramer sequence 5′- rCrGrUrUrGrGrCrArCrCrCrGrArGrArArArUr-UrCrCrArNrNrNrN −3′). RNA was enriched by a third round of binding to Streptavidin M280 beads and TRIzol isolation. RNA was retro-transcribed using RP1 Illumina primer and SuperScript III (Invitrogen) to generate cDNA libraries. Libraries were then amplified using KAPA HiFi Real-Time PCR Library Amplifi-cation Kit (Peqlab) and Illumina primers containing standard TruSeq barcodes. Amplified libraries were subjected to electrophoresis on 2.5% low melting agarose gel and amplicons from 150 to 300 bp were excised, purified from the gel using NucleoSpin Gel and PCR Clean-up kit (Macherey-Nagel) and sequenced on Illu-mina HiSeq 2500 platform (VBCF NGS facility).

**Transcription elongation inhibition with DRB and release.** PHF3 WT, KO and ΔSPOC HEK293T cells were grown in 15 cm dishes and incubated with 100 μM DRB (Sigma) or DMSO for 3.5 h. Cells for time point 0 were harvested immedi-ately. Alternatively, cells were washed twice with PBS and allowed to recover in normal medium for 10, 25, or 40 min before harvesting. Cells were harvested and immediately processed for nuclei isolation as described in the PRO-seq section.

**RNA isolation, RT-qPCR and RNA-seq library preparation.** RNA was isolated from harvested cells using TRI reagent (Sigma) according to the manufacturer's instructions. cDNA was obtained by reverse transcription of 1 μg of RNA using random hexamer primers (Invitrogen) and ProtoScript II Reverse Transcriptase (NEB) according to manufacturers' instructions. RT-qPCR was performed on a BioRad CFX384 Touch qPCR cycler using iTaq Universal Sybr Green Supermix (BioRad). Biorad CFX Maestro software was used to determine Ct values. RT-qPCR data were analyzed by normalizing the expression of the genes of interest by GAPDH housekeeping gene expression; gene expression was calculated after assessing primers efficiency. For RNA-seq, $8 \times 10^6$ HEK293 cells were mixed with $2 \times 10^6$ Drosophila S2 cells for spike-in normalization. Total isolated RNA was first treated with recombinant DNaseI (Roche), cleaned up using peqGOLD PhaseTrap A tubes (Peqlab), and rRNA-depleted using the Ribo-Zero kit (Illumina). RNA-seq libraries were prepared using the NEBNext Ultra II directional RNA library pre-paration kit (NEB) according to the manufacturer's instructions. Sequencing was performed on Illumina HiSeq 2500 (VBCF NGS facility).

**Chromatin immunoprecipitation.** Cells were harvested, counted, resuspended in 50 mL PBS/$10^8$ cells and fixed for 10 min by adding formaldehyde to a final concentration of 1%. Formaldehyde was quenched by adding glycine pH 3 to a final concentration of 0.6 M for 15 min. Cells were centrifuged and washed twice in cold PBS. To isolate nuclei, $10^8$ fixed cells were resuspended in 5 mL cold lysis buffer 1 (50 mM Hepes/KOH pH 7.5, 140 mM NaCl, 1 mM EDTA, 10% glycerol, 0.5% Nonidet P-40, 0.25% Triton X-100, 1x protease inhibitor; for Pol II ChIP 2 mM Na₃VO₄ and 2 mM NaF), rotated for 10 min at 4 °C and centrifuged. Nuclei were resuspended in 5 mL cold lysis buffer 2 (10 mM Tris-Cl pH 8, 200 mM NaCl, 1 mM EDTA, 0.5 mM EGTA, 1x protease inhibitors; for Pol II ChIP 2 mM Na₃VO₄ and 2 mM NaF), rotated for 10 min at room temperature and centrifuged. The pellet was resuspended in 3 mL lysis buffer 3 (10 mM Tris-Cl pH 8, 100 mM NaCl, 1 mM EDTA, 0.5 mM EGTA, 0.1% Na-deoxycholate, 0.5% N-lauroylsarcosine, 1x protease inhibitors; for Pol II ChIP 2 mM Na₃VO₄ and 2 mM NaF). Chromatin was sheared to an average size of 200–600 bp using the Bioruptor Pico (Diagenode) for 20 cycles, 30 s on/30 s off. Triton X-100 was added to a final concentration of 1%. In total 5–10% of chromatin was kept as an input, to the remainder antibody (Pol II pS5 3E8; Pol II pS2 3E10; Pol II pS7 4E12; total Pol II clone F-12 Santa Cruz sc-55492; TCEA1 Abcam ab185947; H3K27me3 Millipore 07–449) or antiserum (GFP Abcam ab290) was added and rotated ON at 4 °C. Antibody and cell amounts are indicated in the key resources table. For TFIIS, Pol II F-12, and H3K27me3 ChIP, chromatin was mixed with 2.5% of mouse chromatin as a spike-in before adding the antibody. Protein G or protein A Dynabeads were washed three times in cold block solution (0.5% BSA in PBS), antibody-bound chromatin was added to the beads and rotated 4–6 h at 4 °C. Beads were washed 5 times (8 times for Pol II pS5 ChIP) in RIPA washing buffer (50 mM Hepes/KOH pH 7.5, 500 mM LiCl, 1 mM EDTA, 1% NP-40, 0.7% Na-deoxycholate) and once in 50 mM NaCl in TE. Crosslinked protein-DNA complexes were eluted in 200 μL elution buffer (50 mM Tris-Cl pH 8, 10 mM EDTA, 1% SDS) for 15 min at 65 °C. Crosslinks were reversed at 65 °C ON. RNA was degraded by adding 0.2 mg/mL RNase A for 2 h at 37 °C, proteins were digested by adding 0.2 mg/mL proteinase K and 5.25 mM CaCl₂ for 30 min at 55 °C. DNA was purified by phenol-chloroform extraction, ethanol-precipitated, and resuspended in 50 μL nuclease-free water. Next generation sequencing libraries were prepared using the NEBNext Ultra II DNA library Prep Kit for Illumina and NEBNext Multiplex Oligos Primer Set 1–3 (New England Biolabs) according to the manufacturer's instructions. Next generation sequencing was performed on Illumina HiSeq 2500, NextSeq 550 or NovaSeq 6000 (VBCF NGS facility). ChIP-qPCR analysis of input and ChIP-DNA was performed on a BioRad CFX384 Touch cycler using Takyon No Rox SYBR MasterMix dTTP Blue (Eurogentec). Data were analyzed by calculating the %input value, values were averaged from three independent experiments.

**Pol II phosphorylation, elongation complex (EC) preparation, and sucrose gradient ultracentrifugation.** Pol II was phosphorylated with DYRK1A kinase as described above. A nucleic acid scaffold for transcribing Pol II was assembled by mixing equimolar amounts of DNA (EC-phf3-template) and RNA (RNA50) in a BioRad T100 Thermal Cycler heated to 95 °C and cooled in 0.1 °C/s increments until 4 °C was reached. For sucrose gradient ultracentrifugation, the Pol II-EC was assembled by incubating 60 pmol Pol II with a 2-fold molar excess of DNA/RNA for 10 min on ice, followed by 10 min at 30 °C, and another 10 min at 30 °C after adding a 4-fold molar excess of non-template DNA (EC-phf3-nontemplate) to generate a transcription bubble. A 4-fold molar excess of PHF3 or TFIIS$^M$/TFIIS$^M$ +TFIIF was incubated with Pol II-EC for 20 min at 25 °C, followed by addition of the 4-fold molar excess of the competitor (TFIIS$^M$/TFIIS$^M$+TFIIF or PHF3 respectively) for 20 min at 25 °C. 10–30% sucrose gradients were prepared using a gradient mixer (Gradient Master 108; BioComp Instruments). Pol II complexes were applied on top of the gradient followed by ultracentrifugation 105169 g in a SW60 swinging bucket rotor (Beckman Coulter) for 16 h at 4 °C. 80 μl fractions were collected carefully from top to the bottom of the tube and analyzed by Western blotting.

**In vitro transcription elongation assay.** Pol II phosphorylation and transcription bubble assembly were performed as above, using arrest sequences comprising a region for EC assembly and a region containing a previously characterized Pol II arrest site shown to be responsive to TFIIS (Arrest-template/5′-FAM-Arrest-RNA and Arrest-nontemplate)[122]. 0.12 μM Pol II was used per reaction. A total of 0.1 μM TFIIS and different concentrations of PHF3 or PHF3 ΔTLD were added to Pol II-EC and incubated for 5 min at 30 °C. Transcription was initiated by adding 100 μM of NTPs in a transcription buffer (20 mM Hepes pH 7.5, 75 mM NaCl, 3 mM MgCl$_2$, and 4% glycerol) and incubating at 37 °C for 10 min. Final ATP concentration was 200 μM due to the leftover from the kinase reaction. Reaction were stopped by adding urea loading buffer (4 M urea in TBE) and EDTA (12.5 mM) and boiling at 95 °C for 5 min. After chilling on ice, samples were incubated with 0.1 mg/mL proteinase K at 37 °C for 20 min, boiled at 95 °C for 5 min and chilled on ice before loading on a 20% denaturing acrylamide gel. Gels were run at 300 V for 1.5 h and scanned using Typhoon (GE Healthcare).

**SLAM-seq.** SLAM-seq was performed as described[50]. HEK293T cells seeded into 6 cm dishes the day before the experiment were incubated in standard culture medium containing 100 μM 4-thiouridine (s$^4$U; ChemGenes) for 12 h with media exchanges every 3 h. Subsequently, cells were washed twice with PBS and incubated with 10 mM uridine-containing medium. Cells were harvested using TRI reagent (Sigma) at timepoints 0 h, 6 h, and 12 h after removal of s$^4$U. RNA was isolated according to the manufacturer's instructions including 0.1 mM DTT during iso-propanol precipitation and resuspended in 1 mM DTT. Isolated RNA was treated with 10 mM iodacetamide to alkylate s$^4$U and subjected to ethanol precipitation. Alkylated RNA was resuspended in water and treated with TURBO DNA-free Kit (Invitrogen) according to the manufacturer's instructions. For global SLAM-seq, libraries were prepared using QuantSeq 3′ mRNA-Seq Library Prep Kit FWD for Illumina (Lexogen) according to the manufacturer's instructions. For targeted analysis of INA, RNA was reverse transcribed using oligo(dT) primer. Subsequently, RNA was removed using RNASe H (NEB) according to the manufacturer's instructions and cDNA was amplified using INA-specific forward primer (5′- CA CGACGCTCTTCCGATCTNNNNNNNTCTGTCCAGCAGTCACTTCG-3′) and oligo(dT) primer (5′-GTTCAGACGTGTGCTCTTCCGATCT-(T)n-V-3′) for 23 cycles. Library amplification was performed using Illumina FWD (5′- AATGATA CGGCGACCACCGAGATCTACACTCTTTCCCTACACGACGCTCTTCCGAT CT-3′) and REV (5′- CAAGCAGAAGACGGCATACGAGATNNNNNNNGTGAC TGGAGTTCAGACGTGTGCTCTTCCGATCT-3′) index primers. Next generation sequencing was performed at the VBCF NGS facility using Illumina NextSeq 550.

**Genomic region definition.** To reliably quantify gene activity, transcription start sites (TSS) and gene body regions were precisely defined. TSS for HEK293 cells were extracted from the FANTOM5 data set (24670764), and transferred from hg19 to the hg38 version of the human reference genome using liftOver (16381938). Regions which mapped to multiple locations were disregarded. Each Fantom TSS was extended into a promoter region of ±250 bp from the putative end of the TSS region. Gene body regions were defined in the following way. Firstly, gene models were downloaded from the Ensembl database (27899575) on 02.10.2015. Each promoter region was assigned to the nearest transcript on the corresponding strand from the Ensembl annotation; promoters that were >2 kb away from a transcript were removed from the data set. Gene body region was defined as a region from the promoter end (+250 bp from the TSS) to the most commonly annotated transcript end for the corresponding gene (a transcript end which is supported by highest number of annotated isoforms). If multiple transcript ends had the same support, the longest isoform was chosen as the representative. If the corresponding transcript overlapped with multiple defined TSS regions, a representative promoter was chosen for each gene by selecting the TSS with the highest average PRO-seq signal.

**Analysis of PRO-seq data.** Prior to quantification of PRO-seq data, genomic regions containing genes were split into promoter regions and gene bodies. PRO-seq data were mapped to the hg38 version of the human reference genome using the STAR- 2.4.0 (23104886) aligner with the following parameters:–outFilterMultimapNmax 10–outFilterMismatchNoverLmax 0.2–sjdbScore 2. PRO-seq signal in the promoter area was quantified by counting the number of read 5′ ends overlapping with the defined promoter regions, while the PRO-seq signal within the gene body was quantified by counting the number of read 5′ ends overlapping the gene body. PRO-seq counts for each region and each sample were normalized to Drosophila S2 spike-in by multiplying the corresponding counts with the ratio between the total number of human and Drosophila reads. Differential analysis was performed using DESeq2[123]. Spike-in calculated normalization factors were normalized to have geometric mean of 1, prior to the differential analysis. To avoid the possibility of quantifying transcriptional read-through from highly expressed genes, all genes which had wild type PRO-seq RPKM > 8 in a 250 bp region 250 bp upstream of the promoter region were filtered out of the analysis. Additionally, all non-expressed genes (genes with promoter RPKM < 2 in all samples) were also filtered from the analysis. PRO-seq data was deposited under the accession number E-MTAB-7501.

**Validation of PRO-seq data.** The quality of PRO-seq data was evaluated using the WT sample, which showed that the most highly expressed genes are microRNAs, histones, Pol II and Jun, as already reported. We compared the expression values in the TSS regions between FANTOM5 data based on Cap Analysis of Gene Expression (CAGE) of mRNAs and our PRO-seq data containing nascent RNAs (Supplementary Fig. 10). We obtained a high Spearman correlation between our data and the CAGE data (0.57) and we observed a spread in the PRO-seq data where CAGE data have low signal, indicating that PRO-seq captures the signal in a much lower dynamic range. In addition, we compared the PRO-seq data with the NET-seq from HEK293 cells for the TSS regions[124]. NET-seq data bigWig files were downloaded from the GEO database (GSE61332), and transferred from hg19 to hg38 genome versions using CrossMap[125]. The NET-seq signals were summarized using the Bioconductor package genomation and compared to the average normalized PRO-seq signal using the Spearman correlation coefficient. The correlation was visualized using ComplexHeatmap[126]. The Spearman correlation between PRO-seq and NET-seq samples was >0.85, which confirms that the data are of high quality. The heatmap shows the correlations between PRO-seq data for PHF3 WT, KO, ΔSPOC, and NET-seq data (Supplementary Fig. 10b).

**Leading edge definition for elongation rate calculation.** PRO-seq DRB time-course data were mapped as described above. Genomic tracks were constructed from the mapped data (in a strand specific manner), and normalized to Drosophila spike-in. Leading edge definition was performed using a heuristic algorithm, which joins regions of increased read density in a strand specific manner. The analysis was performed on difference genomic tracks, where the 0 min time point was subtracted from each subsequent time point. This was done to prevent merging of the signal from genes in head to tail orientation. First, the genome was split into 1 kb non-overlapping tiles, the number of bases covered in each tile was counted, and the coverage in each tile was summed. To smoothen the signal, number of covered bases in each tile was calculated as max(countsn-1, countsn)—a maximum of the current and the preceding tile, in a non-iterative way. Tiles with less than 25% of bases covered, and a minimal normalized signal value of 0.1 (sum of coverage over the tile) were removed from the analysis. From the resulting tiles, a contiguously expressed region was constructed by merging tiles within 4 kb. All regions shorter than 3 kb were filtered out from further analysis. Each contiguously expressed region was annotated to the gene with the closest upstream TSS. Genes with the expressed region defined in all time points for all conditions were kept for further analysis. PRO-seq data from the DRB time-course experiment was deposited under the accession number E-MTAB-8278.

**Analysis of ChIP-seq data.** PHF3, Pol II (F-12), TFIIS and H3K27me3 ChIP-seq data were mapped to the hg38 version of the human genome using Bowtie2[127] with the following parameters: bowtie2 -k 1. TFIIS, Pol II (F-12), and H3K27me3 ChIP-seq samples were normalized using mouse chromatin spike-in and mapped separately to the human (hg38) and mouse (mm10) genomes, using Bowtie2, as implemented in PigX pipeline[128]. The scaling factor was obtained by dividing the total number of uniquely mapped reads to the human genome, with the number of uniquely mapping reads to the mouse genome. The genomics tracks were then constructed by extending the reads to 200 bp into 3′ direction, calculating the coverage vector, and scaling using the aforementioned scaling factor. PHF3 ChIP-seq data was deposited under the accession number E-MTAB-8783. Pol II (F-12), TFIIS and H3K27me3 ChIP-seq data was deposited under the accession number E-MTAB-8789.

**Construction of genomic tracks.** Genomic tracks were constructed by merging all replicates of the corresponding biological conditions (WT, KO, ΔSPOC) and experiments (RNA-seq, PRO-seq, ChIP-seq, SLAM-seq). Merged tracks were then normalized to the total number of reads. For ChIP-seq, the reads were firstly extended to 200 bp towards the 3′ end, the coverage was calculated, and normalized

to the total number of reads. The tracks were additionally normalized by taking the log2 ((ChIP +1) / (Input+1)). Negative values were censored to zero.

**Signal profile construction**. Signal profiles were constructed by averaging the signal from genomic tracks over different functional regions into 100 bins of equal size. The extreme values in the profiles were avoided by applying the trimmed mean function, with the trim parameter set to 0.3. To compare ChIP-seq profiles of different antibodies (different proteins), the profiles were normalized prior to averaging the within region signal range by dividing the signal by min – max.

**Analysis of HEK293T RNA-seq data**. RNA-seq reads were mapped to a genome comprising of human reference genome hg38 version and Drosophila melanogaster version dm6. STAR- 2.5.3 was used with the default parameters and gencode v28 gtf annotation as gtf file. RNA-seq data were quantified using STAR quantMode. Differential expression was analyzed using DESeq2 and the gene counts were normalized to the total Drosophila spike-in counts; genes with an adjusted p-value < 0.05 were designated as differentially expressed. HEK293T RNA-seq data was deposited under the accession number E-MTAB-7498.

**Analysis of mES RNA-seq data**. Mouse embryonic stem cell RNA-seq data were mapped to the mm9 version of the mouse reference genome using STAR- 2.4.0. STAR index was constructed with gene annotation downloaded from the Ensembl database on 20.05.2015. The expression was quantified and the differential expression was analyzed as described previously. mESC RNA-seq data was deposited under the accession number E-MTAB-7526.

**Analysis of SLAM-seq data**. Raw sequenced reads were processed with the SLAMdunk pipeline as previously described[50]. Genes which had detectable conversion rates in at least three biological replicates in all conditions were kept for subsequent analysis. Furthermore, genes which had a non-monotonic decrease of the median conversion rates were filtered out. To estimate the half-lives, a robust linear model was fit on the linearized form of the exponential decay equation using the RLM function from the MASS R package. SLAM-seq data was deposited under the accession numbers E-MTAB-7898 and E-MTAB-7899.

**Sequencing data integration**. The complete data integration and data analysis were done in R using Bioconductor[129], and the following libraries were used: GenomicAlignments[130], data.table (Matt Dowle and Arun Srinivasan, 2017), data.table: Extension of 'data.frame' (R package version 1.10.4-3.), biomaRt[131], GenomicRanges, rtracklayer[132], SummarizedExperiment (10.18129/B9.bioc.SummarizedExperiment), genomation[133], and ggplot2 (10.1007/978-0-387-98141-3).

**Differentiation of mESCs into neural stem cells, neurons, and astrocytes**. mESCs differentiation into neural stem cells (NSCs), and later in neurons and astrocytes, was adapted from a previously described protocol[134]. Briefly, $10^4$ mESCs/cm² were seeded on gelatin-coated 10 cm dishes and cultured for 7 days in N2B27 medium. After 7 days, $2–5 \times 10^6$ cells were transferred to non-gelatinized T75 flasks in NS-N2B27 medium (N2B27 medium supplemented with 10 ng/mL EGF and 10 ng/mL FGF2) and grown for 2–4 days to form aggregates in suspension. The cell aggregates were then collected by centrifugation (105 g for 30 s) and transferred to fresh gelatin-coated T75 flasks and grown in NS-N2B27 medium. After 3 to 7 days, cells displayed NSCs morphology. For neuronal differentiation, NSCs were seeded in NS-N2B27 medium at a density of 25000 cells/cm² on laminin-coated glass coverslips in 24-well plates for immunofluorescence and 6-well plates for RNA isolation. The day after, the medium was replaced with N2B27 medium supplemented with only 5 ng/mL FGF2. Cells grown on glass coverslips were then fixed for immunofluorescence, while cells grown in 6-well plates were harvested for RNA isolation at the indicated time points. Since cell quantification was not possible due to the organization of WT differentiated cells into tight aggregates, to quantify the differences between Phf3 WT and KO cells upon neuronal differentiation we manually counted by fluorescence microscopy all TuJ1 positive cell aggregates (referred to as "neuronal clumps") on the glass coverslips. For astrocyte differentiation, NSCs were seeded in N2B27 medium supplemented with 1% FBS at a density of 50000 cells/cm² on gelatin coated glass coverslips in 24-well plates. Cells were fixed after 5 days and samples were processed for immunofluorescence.

**Exit from naive pluripotency assay of mESCs**. mESCs were cultured in N2B27 medium (DMEM F12 + Neurobasal medium supplemented with 1% L-glutamine, 1% penicillin/streptomycin, 1% NEAA, B27 supplement, N2 supplement, 2-mercaptoethanol) supplemented with 2i/LIF as previously described[102] for at least two passages. $10^4$ cells were seeded in 6-well plates and grown for 36 h in N2B27medium + 2i in the presence or absence of LIF. Subsequently, cells that were grown without LIF were incubated in the absence of 2i to allow differentiation for 8 h and 24 h. After harvesting the cells, RNA was isolated to determine gene expression by RT-qPCR.

**Quantification and statistical analysis**. Error bars represent standard deviation estimated from three to four independent experiments. For Western blot band intensity, EU fluorescence intensity and RT-qPCR analyses statistical significance was calculated using one-tailed Student's t-test. P-values smaller than 5% were considered statistically significant and indicated with an asterisk ('*' for $p < 0.05$; '**' for $p < 0.01$; '***' for $p < 0.001$; '****' for $p < 0.0001$). ChIP-seq, RNA-seq and PRO-seq were performed in triplicates. SLAM-seq was performed in six replicates. Statistical analysis on all sequencing data was performed using DESeq2. Statistical significance of the differential expression/abundance was determined by two-tailed Wald test, after appropriate normalization for each type of data.

**Reporting summary**. Further information on research design is available in the Nature Research Reporting Summary linked to this article.

## Data availability
The data that support this study are available from the corresponding author upon reasonable request. The source data are provided with this paper. The atomic coordinates for the structures generated in this study have been deposited in the Protein Data Bank under accession codes: 6IC8 for PHF3 SPOC:2xpS2, 6IC9 for PHF3 SPOC:2xpS2pS7, 6Q2V for PHF3 SPOC, 6Q5Y for PHF3 SPOC:2xpS2pS5. The sequencing data generated in this study have been deposited in ArrayExpress under accession codes: E-MTAB-7498 (RNA-seq HEK293T), E-MTAB-8783 (PHF3), E-MTAB-8789 (Pol II F-12, TFIIS, H3K27me3), E-MTAB-7501 (PRO-seq), E-MTAB-8278 (Pro-seq elongation rate), E-MTAB-7898 and E-MTAB-7899 (SLAM-seq), E-MTAB-7526 (RNA-seq mESC). The mass spectrometry proteomics data generated in this study have been deposited in the ProteomeXchange Consortium via the PRIDE partner repository under accession code PXD026292. The processed mass spectrometry and sequencing data are provided in Supplementary Data 1–6. All other raw data generated in this study are provided in Supplementary Data 7. Atomic coordinates used in this study are available in the Protein Data Bank under accession codes 2RT5, 4BY7, 5KXF, 5IYB, 6GMH, 6IC8. The NET-seq data used in this study are available in GEO under accession code GSE61332. The ATAC-seq data used in this study are available in ArrayExpress under accession code E-MTAB-6195. H3K4me3 ChIP-seq data used in this study are available from ENCODE under accession code ENCSR000DTU. REST ChIP-seq data used in this study are available from ENCODE under accession code ENCSR896UBV.

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

## Acknowledgements

D.S. thanks Claudine Kraft, Renée Schroeder, Verena Jantsch, Franz Klein and Peter Schlögelhofer for support. We thank Anita Testa Salmazo for help with purifying Pol II; Matthias Geyer and Robert Düster for sharing DYRK1A kinase; Felix Hartmann and

Clemens Plaschka for help with mass photometry; Goran Kokic for design of the arrest assay sequences; Petra van der Lelij for help with generating mESC KO; Maximilian Freilinger for help with the purification of mEGFP-CTD; Stefan Ameres, Nina Fasching and Brian Reichholf for advice on SLAM-seq and for sharing reagents; Laura Gallego Valle for advice regarding LLPS assays; Krzysztof Chylinski for advice regarding CRISPR/Cas9 methodology; VBCF Protein Technologies facility for purifying PHF3 and providing gRNAs and Cas9; VBCF NGS facility for sequencing; Monoclonal antibody facility at the Helmholtz center for Pol II antibodies; Friedrich Propst and Elzbieta Kowalska for advice and for sharing materials; Egon Ogris for sharing materials; Martin Eilers for recommending a ChIP-grade TFIIS antibody; Susanne Opravil, Otto Hudecz, Markus Hartl and Natascha Hartl for mass spectrometry analysis; staff of the X-ray beamlines at the ESRF in Grenoble for their excellent support; Christa Bücker, Anton Meinhart, Clemens Plaschka and members of the Slade lab for critical comments on the manuscript; Life Science Editors for editing assistance. M.B. and D.S. acknowledge support by the FWF-funded DK 'Chromosome Dynamics'. T.K. is a recipient of the DOC fellowship from the Austrian Academy of Sciences. U.S. is supported by the L'Oreal for Women in Science Austria Fellowship and the Austrian Science Fund (FWF T 795-B30). M.L is supported by the Vienna Science and Technology Fund (WWTF, VRG14-006). R.S. is supported by the Czech Science Foundation (15-17670 S and 21-24460 S), Ministry of Education, Youths and Sports of the Czech Republic (CEITEC 2020 project (LQ1601)), and the European Research Council (ERC) under the European Union's Horizon 2020 research and innovation programme (Grant agreement no. 649030); this publication reflects only the author's view and the Research Executive Agency is not responsible for any use that may be made of the information it contains. M.S. is supported by the Czech Science Foundation (GJ20-21581Y). K.D.C. research is supported by the Austrian Science Fund (FWF) Projects I525 and I1593, P22276, P19060, and W1221, Federal Ministry of Economy, Family and Youth through the initiative 'Laura Bassi Centres of Expertise', funding from the Centre of Optimized Structural Studies No. 253275, the Wellcome Trust Collaborative Award (201543/Z/16), COST action BM1405 Non-globular proteins - from sequence to structure, function and application in molecular physiopathology (NGP-NET), the Vienna Science and Technology Fund (WWTF LS17-008), and by the University of Vienna. This project was funded by the MFPL start-up grant, the Vienna Science and Technology Fund (WWTF LS14-001), and the Austrian Science Fund (P31546-B28 and W1258 "DK: Integrative Structural Biology") to D.S.

## Author contributions

L.A. generated endogenous PHF3 GFP-tagged and SPOC-deleted cell lines, performed all ChIP and SLAM-seq experiments, purified SPOC and Pol II, performed and analyzed FCS experiments; V.F. designed and performed the analysis of all sequencing data and conceptually drove the project; M.B. generated PHF3 KO, performed and analyzed co-immunoprecipitation, EU incorporation, neuronal differentiation experiments, performed RNA-seq and PRO-seq, generated PHF3 KO cell lines stably expressing PHF3 and ΔSPOC PHF3 and performed complementation experiments; I.G. solved the SPOC structures; A.K. performed and analyzed FA experiments; T.K. performed and analyzed Airyscan imaging and FRAP; U.S. performed PRO-seq experiments; M.P. supervised and analyzed FCS experiments; S.K. purified SPOC; C.E. performed initial in vitro LLPS assays; M.S. provided mEGFP-CTD plasmid and advice on in vitro LLPS assays; E.B. analyzed mass spectrometry data; K.M. supervised mass spectrometry analysis; G.L. and A.V. analyzed HEK293 RNA-seq data; M.L. supervised ESC genome-engineering and differentiation experiments; R.P. supervised initial Pol II ChIP experiments; A.S. supervised the analysis of HEK293 RNA-seq data; A.A. designed and supervised the analysis of sequencing data; R.S. supervised FA experiments; C.B. supervised Pol II in vitro assays; K.D.C. supervised X-ray analysis; D.S. conceived the study, performed, supervised, analyzed experiments, and wrote the manuscript.

## Competing interests

The authors declare no competing interests.
