## [Peer Review File · Nature Communications]

PHF3 regulates neuronal gene expression through the Pol II CTD reader domain SPOCREVIEWER COMMENTS

Reviewer #1 (Remarks to the Author):

The manuscript by Lisa-Marie Appel and co-workers describes the crystal structure of the SPOC domain of human PHF3 in complex with the phosphorylated C-terminal domain (CTD) of RNA polymerase II. The authors determine several crystal structures of differently phosphorylated tandem-heptarepeat peptides of the CTD bound to the SPOC domain to high resolution, of which the pS2 mark in two adjacent hepta-repeats binds best. The PHF3 protein, but not the SPOC domain alone, contributes to liquid-liquid phase separation (LLPS) of S2 phosphorylated CTD, forming large condensates. The authors find that PHF3 competes with its TLD domain (TFIIS-like domain) with the transcription factor TFIIS for the binding to Pol II, thereby inhibiting the rescue of backtracked Pol II. The TLD is directly preceding the SPOC domain, forming an assembly that associates and travels with elongating Pol II phosphorylated on Ser2. The authors observe that PHF3 loss or SPOC domain deletion correlates with an increase in mature transcripts and show by SLAM-seq that PHF3 negatively regulates mRNA stability. Gene ontology analysis with CRISPR/Cas9 generated PHF3 knock-out cells (HEK293T) reveals that PHF3 negatively regulates neuronal gene expression and differentiation. Overall, the authors establish PHF3 as a new factor regulating transcription elongation and mRNA stability with a particular emphasis on neuronal gene expression.

This is an elegant and very comprehensive study that reaches out from structural biology and transcriptomics to LLPS biochemistry and neuronal differentiation in embryonic stem cells. The findings from this study bring a new CTD domain onto the tableau of transcriptional regulators, which I suppose will be very interesting to many researchers in the field. The manuscript is well written, the data are clearly presented, and the figures are concise. I only have a couple of comments, critical considerations, and suggestions to this study.

Criticism

For all doubly phosphorylated CTD hepta-repeats the dissociation constants of binding to the wild-type PHF3 SPOC domain varies only between 0.8 and 48 μM . While pS2pS7 with 0.8 μM and pS2 with 1.6 μM bind best, pS5 with 20 μM or pS7 with 26 μM show at least a 10-fold difference. A consecutive phosphorylation mark in two following repeats seems thus mandatory for binding and the main selection criteria. Is anything known if PHF3 dimerizes such that SPOC domain recognition of pCTD could be synergistic? Please comment.

It has been previously shown that phosphorylation of the CTD abrogates its accumulation in LLPS condensates. Here the authors show that full length PHF3 forms droplets and interestingly, that PHF3 is also able to rescue S2 phosphorylated CTD into droplets. This is mechanistically very interesting, as the pS2-SPOC domain interaction in LLPS could change the properties of the CTD in such condensates and shift the stability of the dissolution of the condensates upon phosphorylation in a concentration dependent manner. Maybe, the author could add such perspectives into the discussion section.

This reviewer suggests to delete the word "new" from the title of the manuscript. Although the identification of SPOC domains as RNA pol II CTD binders is new, it is sufficient to describe this novelty as it is currently in the second sentence of the abstract "as a new regulator of transcription ...".

It might be helpful to include the cartoon model of the proposed PHF3-mediated backtracking mechanism by its competition with TFIIS from the Extended Data Fig. 11b into the main text of the manuscript. It could be shown potentially in Fig. 5.

Reviewer #2 (Remarks to the Author):

In this manuscript, Appel and colleagues structurally and functionally characterize the relatively unexplored transcriptional regulator PHF3. The authors discovered that the SPOC domain of PHF3 binds to the phosphorylated C-terminal domain (CTD) of RNA polymerase II (Pol II). The authors also show that PHF3 colocalizes with Pol II clusters inside cells and forms droplets capable of absorbing phosphorylated CTD and Pol II. Using a combination of cell biology assays the authors were able to define the biological significance of this interaction, demonstrating that PHF3 plays important roles in Pol II-dependent transcription and mRNA stability. Overall, this manuscript describes novel findings, contains excellent quality data, and has justifiable conclusions.

A few minor comments:

1. Labels of doubly and triply phosphorylated peptides in the text, figures, and tables are inconsistent and confusing.
2. There is no rationale included to obtain the structure with the pS7-containing diheptapeptide. What does this interaction mean for biology? How the structures of the SPOC domain with pS2, pS5 and pS7 help to understand the biological role of PHF3?
3. While the authors compare the structures of the SPOC domain bound by the pS2 and pS2pS7 diheptapeptides, a similar comparison is lacking for pS2pS5.
4. The curves shown in Fig 2b and c likely require two-site binding analysis.
5. Unclear the relationship between the abilities of full length PHF3 and separately phosphorylated CTD to form droplets and the reduction of the self-association propensity of PHF3 and the reduction of the size of the droplets.
6. Formation of the PHF3 'droplets' needs to be confirmed by FRAP.

Reviewer #3 (Remarks to the Author):

In the present study, Appel et al. use a combination of mainly biochemistry and functional genomics approaches to study the functional role of the PHD-finger protein 3 (PHF3) in human cells and mouse embryonic stem cells (including differentiating cells). The authors identify PHF3 as a direct interactor of RNA polymerase II (Pol II) and convincingly show that this interaction is mediated through the SPOC domain of PHF3 that preferentially binds to the Ser2-phosphorylated CTD of Pol II. Using ChIP-seq the authors provide strong evidence that PHF3 co-localizes with CTD phosphorylated forms of Pol II along genes suggesting that PHF3 is recruited at an early stage during transcription and then travels with Pol II across the gene-body. The authors applied CRISPR-Cas9 gene editing to generate several new cell lines including cell lines in which either PHF3 or the SPOC domain were deleted. By using RNA-seq the authors find that upon both deletions steady-state RNA levels for a set of transcripts were increased. Remarkably, the authors provide insights into the mechanism of Pol II transcription regulation by PHF3. Using several orthogonal approaches Appel et al. shows that PHF3 competes with TFIIS for Pol II binding and can displace TFIIS from Pol II elongation complexes. The authors further provide evidence that PHF3 is implicated in the regulation of RNA stability by providing clear evidence that mRNA half-lives are increased upon PHF3 and SPOC deletions. Finally, the authors demonstrate that PHF3 is involved in the regulation of neuronal gene expression and is required for neuronal cell differentiation.

I am impressed by this study and the broad spectrum of experimental approaches that the authors

have used to reveal new roles of PHF3 in Pol II transcription and RNA stability regulation. A clear strength of this work is the identification and characterization of the direct link between PHF3 and Pol II. The authors uncover the SPOC domain as a new CTD reader domain. The most exciting finding at least for me is that PHF3 represents a molecular link between Pol II transcription and RNA stability regulation. Overall the manuscript is excellently written and main conclusions are supported by the data that is provided. This study also opens up interesting future research directions such as to further characterize the molecular links and regulatory mechanisms between transcription elongation and RNA stability control. I have no doubts that the findings of this study will be of broad interest for researchers in the transcription, RNA and differentiation (particularly neuronal differentiation) communities.

Here are my comments that I hope the authors will find helpful to further improve an already strong manuscript:

Major comments:

(1) The finding that PHF3 or SPOC deletions changed nascent transcript levels at only 68 or 78 genes is puzzling. If PHF3 generally competes with TFIIS for Pol II binding and pause/arrest release I would have expected a stronger impact on nascent transcript levels. How can this be explained? This could be addressed in the discussion section.

The overall impact of PHF3 and SPOC deletions on Pol II transcription is shown in Ext. Fig. 9a and Ext. Fig. 10a. I suggest to move Ext. Fig. 9a to Fig. 4. Levels of transcribing Pol II clearly decrease over the gene-body and the termination zone. Is this decrease significant? Box-plot quantifications would be helpful. In the example gene (GPR50) shown in Fig 4i Pol II signals in the promoter-proximal region are strongly increased upon PHF3 and SPOC deletion. This is consistent with the Pol II ChIP-seq data (Ext. Fig. 10b, PHF3 deletion). I bring this up because this observation along with the finding that Pol II levels are generally decreased over the gene-body would be indicative for impaired pause release which would be in conflict with the claim that 'PHF3 negatively regulates gene expression'. A clarification would be helpful.

(2) The authors provide evidence that PHF3 plays a role in the regulation of RNA stability but the mechanism remains unclear. In order to gain insights into the underlying mechanism authors could re-analyze their PHF3 IP-MS data. Are known regulators of the RNA decay pathway among the interactors? The authors can address this with the data that they already have.

(3) The authors state 'TFIIS ChIP-seq experiments showed increased TFIIS occupancy in PHF3 KO cells'. However, only TFIIS ChIP-seq data for two genes (GPR50 and GAPDH) are shown (Fig. 5f). Can the increase of TFIIS occupancy also be observed at other genes? What fraction of genes? A more global computational analysis would be helpful. Results of this computational analysis can be added to Fig. 5. Since no spike-in controls were used in the ChIP-seq experiments I suggest to soften this claim.

Minor comments:

Introduction:

(1) It would be helpful for the reader to provide more information on known cellular functions of PHF3 in the introduction section. At the moment the authors only mention the family to which PHF3 belongs and then provide a more detailed description of the domain architecture.

(2) Second paragraph, page 2: The authors provide a very nice overview of CTD interactors. It would be helpful if the authors could add clarification on whether these interactions refer to CTD Ser2-P only and indicate for which organism these interactions were detected. I guess that most of these studies were performed in yeast. Similarly, on pages 13 (Discussion section), second paragraph, please clarify

if the listed CTD reader domains that are highlighted refer to Ser2-P binding only and to indicate the organisms.

Results section:

(3) It is not clear to me why the authors performed the PHF3 IP-MS experiments for the overexpression FLAG-PHF3 line and have not used their cell line with endogenous GFP-tagged PHF3. A clarification would be helpful.

(4) Page 4: 'PHF3 SPOC did not bind the unphosphorylated CTD (...) but phosphorylation of S2 within both repeats was sufficient to confer strong binding.' How strong was the binding μ M or nM range? Adding values, if available, would be helpful.

(5) Page 10: 'We observed a significant overall increase in mRNA half-lives (...)'. What is the median increase in min or hours upon PHF3 or SPOC deletions? Please add this information to the main text.

(6) How is the correlation between SLAM-seq and PRO-seq replicate measurements? Adding correlation coefficients of replicate measurements to the main text or a statement to indicate the degree of correlation would clarify this.

Discussion section:

(7) Page 13: 'While the Pol II CTD is the primary anchoring point for PHF3 (...)'. I agree that based on the data presented the CTD plays an important role in PFH3 localization to active transcription units, but CTD-independent interactions with Pol II or with elongation factors might be equally important in cells. Therefore, I suggest to soften this claim.

(8) Minor comments regarding Figure panels:

Fig. 5a and b is quite complex.

Fig. 6a: can both curves been presented in the same plot?

Fig. 6e, f: please add labels for wt, dPHF3, dSPOC

Fig. 7f: GO terms are hard to read. Increasing font sizes would be helpful.

REVIEWER COMMENTS

Reviewer #1 (Remarks to the Author):

The manuscript by Lisa-Marie Appel and co-workers describes the crystal structure of the SPOC domain of human PHF3 in complex with the phosphorylated C-terminal domain (CTD) of RNA polymerase II. The authors determine several crystal structures of differently phosphorylated tandem-heptarepeat peptides of the CTD bound to the SPOC domain to high resolution, of which the pS2 mark in two adjacent hepta-repeats binds best. The PHF3 protein, but not the SPOC domain alone, contributes to liquid-liquid phase separation (LLPS) of S2 phosphorylated CTD, forming large condensates. The authors find that PHF3 competes with its TLD domain (TFIIS-like domain) with the transcription factor TFIIS for the binding to Pol II, thereby inhibiting the rescue of backtracked Pol II. The TLD is directly preceding the SPOC domain, forming an assembly that associates and travels with elongating Pol II phosphorylated on Ser2. The authors observe that PHF3 loss or SPOC domain deletion correlates with an increase in mature transcripts and show by SLAM-seq that PHF3 negatively regulates mRNA stability. Gene ontology analysis with CRISPR/Cas9 generated PHF3 knock-out cells (HEK293T) reveals that PHF3 negatively regulates neuronal gene expression and differentiation. Overall, the authors establish PHF3 as a new factor regulating transcription elongation and mRNA stability with a particular emphasis on neuronal gene expression.

This is an elegant and very comprehensive study that reaches out from structural biology and transcriptomics to LLPS biochemistry and neuronal differentiation in embryonic stem cells. The findings from this study bring a new CTD domain onto the tableau of transcriptional regulators, which I suppose will be very interesting to many researchers in the field. The manuscript is well written, the data are clearly presented, and the figures are concise. I only have a couple of comments, critical considerations, and suggestions to this study.

Thank you very much for your positive feedback!

Criticism

For all doubly phosphorylated CTD hepta-repeats the dissociation constants of binding to the wild-type PHF3 SPOC domain varies only between 0.8 and 48 μM . While pS2pS7 with 0.8 μM and pS2 with 1.6 μM bind best, pS5 with 20 μM or pS7 with 26 μM show at least a 10-fold difference. A consecutive phosphorylation mark in two following repeats seems thus mandatory for binding and the main selection criteria. Is anything known if PHF3 dimerizes such that SPOC domain recognition of pCTD could be synergistic? Please comment.

To address this, we performed mass photometry measurements, which revealed that PHF3 forms monomers, at least *in vitro*. We have included these results in Extended Data Fig. 3.

It has been previously shown that phosphorylation of the CTD abrogates its accumulation in LLPS condensates. Here the authors show that full length PHF3 forms droplets and interestingly, that PHF3 is also able to rescue S2 phosphorylated CTD into droplets. This is mechanistically very interesting, as the pS2-SPOC domain interaction in LLPS could change the properties of the CTD in such condensates and

shift the stability of the dissolution of the condensates upon phosphorylation in a concentration dependent manner. Maybe, the author could add such perspectives into the discussion section.

We have now added this interesting perspective into the Discussion.

‘LLPS of unphosphorylated CTD may facilitate Pol II clustering and transcriptional bursting during transcription initiation, whereas CTD phosphorylation dissolves ‘initiation clusters’ and drives Pol II clustering with RNA processing factors²⁹. Our findings that PHF3 condensates capture phCTD and phPol II suggest that PHF3 may act as a bridge between pS2 CTD and RNA processing factors.’

This reviewer suggests to delete the word “new” from the title of the manuscript. Although the identification of SPOC domains as RNA pol II CTD binders is new, it is sufficient to describe this novelty as it is currently in the second sentence of the abstract “as a new regulator of transcription ...”.

‘New’ is now removed from the title.

It might be helpful to include the cartoon model of the proposed PHF3-mediated backtracking mechanism by its competition with TFIIS from the Extended Data Fig. 11b into the main text of the manuscript. It could be shown potentially in Fig. 5.

We moved the model to the main figures (new Fig. 7i).

Reviewer #2 (Remarks to the Author):

In this manuscript, Appel and colleagues structurally and functionally characterize the relatively unexplored transcriptional regulator PHF3. The authors discovered that the SPOC domain of PHF3 binds to the phosphorylated C-terminal domain (CTD) of RNA polymerase II (Pol II). The authors also show that PHF3 colocalizes with Pol II clusters inside cells and forms droplets capable of absorbing phosphorylated CTD and Pol II. Using a combination of cell biology assays the authors were able to define the biological significance of this interaction, demonstrating that PHF3 plays important roles in Pol II-dependent transcription and mRNA stability. Overall, this manuscript describes novel findings, contains excellent quality data, and has justifiable conclusions.

Thank you very much for your positive feedback!

A few minor comments:

1. Labels of doubly and triply phosphorylated peptides in the text, figures, and tables are inconsistent and confusing.

We have now made sure that CTD peptides are uniformly labelled. We also included a table with all the peptides (new Supplementary Table 1) and we explained that the peptides that were used for the binding assays and X-ray structure analysis were the same except for the FAM label at the N-terminus for those used in the binding assays. In the figures we indicated only the part of the peptide that was visible in the structure.

2. There is no rationale included to obtain the structure with the pS7-containing diheptapeptide. What does this interaction mean for biology? How the structures of the SPOC domain with pS2, pS5 and pS7 help to understand the biological role of PHF3?

We have now included the rationale for obtaining structures with pS7 CTD. The initial reason was a very similar affinity of PHF3 SPOC towards 2xpS2 and 2xpS2pS7, which is why we wanted to understand the mode of binding. Some CTD reader domains were shown to engage two different phospho marks. For example, Rtt103 CID binds pS2pS7 marks (PMID: 29073019), while PIN1 WW domain binds pS2pS5 (PMID: 10932246). The structure with 2xpS2pS7 revealed that PHF3 SPOC engages only pS2 in adjacent repeats, while pS7 is projected away from the SPOC surface. This indicates that PHF3 SPOC only recognizes tandem pS2 marks as a mark of transcription elongation and thereby interacts with the actively elongating Pol II.

‘The three different 2xpS2-containing peptides with comparable binding affinities were used for structural analysis to determine whether PHF3 SPOC exclusively binds tandem pS2 marks or whether it can concomitantly engage pS7 or pS5 as shown for yeast Rtt103 (binds pS2pS7) and human PIN1 (binds pS2pS5)^{44, 45}.’

3. While the authors compare the structures of the SPOC domain bound by the pS2 and pS2pS7 diheptapeptides, a similar comparison is lacking for pS2pS5.

The description of the SPOC:pS2pS5 structure was included in the Supplementary file, the reason being that this structure shows the binding of two SPOC domains to one CTD peptide, whereas FCS measurements revealed 1:1 stoichiometry. Therefore we concluded that this structure most likely represents an artefact of crystal packing and was not discussed in the main text.

From Supplementary:

‘The crystal structure of 2xpS2pS5 CTD peptide bound to PHF3 SPOC showed a different binding mode compared to 2xpS2 and 2xpS2pS7 CTD peptides (Extended Data Fig. 2i-m). Two molecules of SPOC were bound to 2xpS2pS5 peptide, the conformation of the bound peptide was slightly different and three phosphorylated CTD residues (pS5 from the first heptapeptide and pS2pS5 from the second) were forming hydrogen bonds with SPOC residues (Extended Data Fig. 2i-m). SPOC residues K1267 and K1309 from the SPOC_C molecule and R1297 from SPOC_A formed hydrogen bonds with pS5a; R1248 from SPOC_C with pS2b; and R1248 and R1295 from SPOC_C and K1260 from SPOC_A hydrogen bonded with pS5b) (Extended Data Fig. 2m). In sum, both positively charged patches from two SPOC molecules contributed interchangeably to the docking of the three phosphorylated serines (Extended Data Fig. 2m). While 2xpS2 structures indicate that the positive patches on the SPOC surface are geared to accommodate pS2 marks on adjacent repeats, 2xpS2pS5 structure reveals that the binding site can also adjust itself towards binding of phosphomarks within the same repeat. However, the FCS measurements showed a 1:1 binding stoichiometry for both 2xpS2 and 2xpS2pS5 peptides (Extended Data Fig. 3), confirming the data from the crystal structure of 2xpS2 and 2xpS2pS7. This indicates that 2:1 stoichiometry (SPOC:peptide) observed in the 2xpS2pS5 co-structure is most likely an artefact of crystal packing.’

4. The curves shown in Fig 2b and c likely require two-site binding analysis.

The FA measurements with a CTD peptide containing only one phosphorylation site showed no binding (the K_d could not be determined). A two-site binding model assumes two distinct (non-cooperative) binding events, where it should be possible to determine the binding affinity for a single binding event. This is why we analysed our data with a one-site binding model. We have now reanalysed all the data with a two-site binding model (Extra Figure 1). If the reviewer agrees, we would keep the one-site binding analysis as in the original manuscript.

Extra Figure 1. Fluorescence anisotropy (FA) measurement of the binding of **a**, 2xpS2, **b**, 2xpS2pS7, **c**, pS2 on one CTD repeat and non-phosphorylated CTD, **d**, 2xpS2pS5, **e**, 2xpS5, **f**, 2xpS7, **g**, 2xpS5pS7 CTD peptides to PHF3 SPOC WT (black) or R1248A mutant (red). **h**, Fluorescence anisotropy measurement of the binding of 2xpS2 to PHF3 SPOC WT at 25 mM NaCl (red) or 100 mM NaCl (black). Normalized fluorescence anisotropy is plotted as a function of protein concentration ($n=3$). The data were normalized for visualization purposes and the experimental isotherms were fitted to two-site binding model.

5. Unclear the relationship between the abilities of full length PHF3 and separately phosphorylated CTD to form droplets and the reduction of the self-association propensity of PHF3 and the reduction of the size of the droplets.

We have now included additional experiments in the manuscript that show how PHF3 condensation is driven through hydrophobic interactions, while PHF3-phCTD condensates are modulated through electrostatic interactions (Extended Data Fig. 5).

‘PHF3 condensates were larger at a higher salt concentration, confirming that PHF3 LLPS is primarily driven by hydrophobic interactions (Extended Data Fig. 5a,b). Phosphorylated CTD partitioned into PHF3 condensates and modulated their size in a salt-dependent manner (Fig. 3a,b and Extended Data Fig. 5a,b). PHF3-phCTD condensate size decreased with higher salt concentration due to interference with electrostatic interactions between PHF3 SPOC and phCTD (Extended Data Fig. 5a,b). Our results suggest that PHF3 condensates are primarily formed through hydrophobic interactions, while PHF3-phCTD condensation is additionally modulated through electrostatic interactions.’

6. Formation of the PHF3 ‘droplets’ needs to be confirmed by FRAP.

We have now performed FRAP experiments to monitor PHF3 recovery after bleaching the inner part of droplets consisting of PHF3 without or with phCTD or phPol II (Extended Data Fig. 5). PHF3 showed fastest recovery within PHF3-phPol II droplets (half-time~30 s) and slowest recovery within PHF3-phCTD droplets (half-time~150 s). We have also replaced the term ‘droplets’ with ‘condensates’.

Reviewer #3 (Remarks to the Author):

In the present study, Appel et al. use a combination of mainly biochemistry and functional genomics approaches to study the functional role of the PHD-finger protein 3 (PHF3) in human cells and mouse embryonic stem cells (including differentiating cells). The authors identify PHF3 as a direct interactor of RNA polymerase II (Pol II) and convincingly show that this interaction is mediated through the SPOC domain of PHF3 that preferentially binds to the Ser2-phosphorylated CTD of Pol II. Using ChIP-seq the authors provide strong evidence that PHF3 co-localizes with CTD phosphorylated forms of Pol II along genes suggesting that PHF3 is recruited at an early stage during transcription and then travels with Pol II across the gene-body. The authors applied CRISPR-Cas9 gene editing to generate several new cell lines including cell lines in which either PHF3 or the SPOC domain were deleted. By using RNA-seq the authors find that upon both deletions steady-state RNA levels for a set of transcripts were increased. Remarkably, the authors provide insights into the mechanism of Pol II transcription regulation by PHF3. Using several orthogonal approaches Appel et al. shows that PHF3 competes with TFIIS for Pol II binding and can displace TFIIS from Pol II elongation complexes. The authors further provide evidence that PHF3 is implicated in the regulation of RNA stability by providing clear evidence that mRNA half-lives are increased upon PHF3 and SPOC deletions. Finally, the authors demonstrate that PHF3 is involved in the regulation of neuronal gene expression and is required for neuronal cell differentiation.

I am impressed by this study and the broad spectrum of experimental approaches that the authors have used to reveal new roles of PHF3 in Pol II transcription and RNA stability regulation. A clear strength of this work is the identification and characterization of the direct link between PHF3 and Pol II. The authors uncover the SPOC domain as a new CTD reader domain. The most exciting finding at least for me is that PHF3 represents a molecular link between Pol II transcription and RNA stability regulation. Overall the

manuscript is excellently written and main conclusions are supported by the data that is provided. This study also opens up interesting future research directions such as to further characterize the molecular links and regulatory mechanisms between transcription elongation and RNA stability control. I have no doubts that the findings of this study will be of broad interest for researchers in the transcription, RNA and differentiation (particularly neuronal differentiation) communities.

Thank you very much for your positive feedback!

Here are my comments that I hope the authors will find helpful to further improve an already strong manuscript:

Major comments:

(1) The finding that PHF3 or SPOC deletions changed nascent transcript levels at only 68 or 78 genes is puzzling. If PHF3 generally competes with TFIIS for Pol II binding and pause/arrest release I would have expected a stronger impact on nascent transcript levels. How can this be explained? This could be addressed in the discussion section.

We have addressed this point in the Discussion section.

‘Why do genes react differently to PHF3 loss? Neuronal genes were enriched among de-repressed genes with high levels of nascent and/or mature transcripts. These genes are low-expressed and Polycomb-repressed in WT cells but bear the marks of ‘poised’ Pol II (H3K4me3, Pol II pS5). Poised Pol II is found in ESCs as well as differentiating and post-mitotic cells. During neuronal differentiation, poised Pol II primes neuronal transcription factors for activation whilst keeping non-neuronal genes silenced. PHF3 may prevent efficient TFIIS-mediated rescue from backtracking and induce premature termination. Reactivation of these genes would thus be highly dependent on TFIIS, which may be the reason for their marked derepression in PHF3 KO cells where TFIIS would gain more access to Pol II.’

The overall impact of PHF3 and SPOC deletions on Pol II transcription is shown in Ext. Fig. 9a and Ext. Fig. 10a. I suggest to move Ext. Fig. 9a to Fig. 4. Levels of transcribing Pol II clearly decrease over the gene-body and the termination zone. Is this decrease significant? Box-plot quantifications would be helpful.

We moved Extended Data Fig. 9 to new Fig. 5a. We added box plots to Extended Data Fig. 8c,d. We also performed statistical analysis using Wilcoxon signed-rank test, which showed that the differences between WT and KO or Δ SPOC are statistically significant for all conditions (not only gene body and pA), which is not surprising considering that this analysis was based on ~30 000 genes. However, when all genes are analysed, the size effects are very small (Extended Data Fig. 8c), while a subset of RNA-seq upregulated genes in KO and Δ SPOC showed pronounced size effects (Extended Data Fig. 8d).

In the example gene (GPR50) shown in Fig 4i Pol II signals in the promoter-proximal region are strongly increased upon PHF3 and SPOC deletion. This is consistent with the Pol II ChIP-seq data (Ext. Fig. 10b, PHF3 deletion). I bring this up because this observation along with the finding that Pol II levels are generally decreased over the gene-body would be indicative for impaired pause release which would be in conflict with the claim that ‘PHF3 negatively regulates gene expression’. A clarification would be helpful.

We have now calculated stalling index (TSS/gene body) for PRO-seq and Pol II (F-12) ChIP-seq data, taking into account ‘all genes’ or only ‘RNA-seq UP in PHF3 KO and Δ SPOC’. Indeed, we found that stalling index is increased based on Pol II ChIP-seq analysis for both gene sets and based on PRO-seq

analysis for RNA-seq upregulated genes. This would indicate that PHF3 exerts a positive effect on pause release. Given that a change in Pol II distribution is also indicative of a change in elongation rate, we performed a DRB time course experiment, which revealed a reduction in early elongation rate in PHF3 KO and Δ SPOC. Therefore, PHF3 seems to act globally as a positive regulator of pause release and transcription elongation, although its loss has little effect on nascent transcript levels. However, a small subset of ~70 genes showed a pronounced increase in nascent transcription and increased TFIIS occupancy in PHF3 KO and Δ SPOC. Therefore, globally PHF3 seems to act as a positive regulator of transcription elongation and a negative regulator of RNA stability, whereas on a small subset of genes PHF3 acts as a negative regulator of both transcription and RNA stability. We have added this analyses to the manuscript as new Fig. 5.

(2) The authors provide evidence that PHF3 plays a role in the regulation of RNA stability but the mechanism remains unclear. In order to gain insights into the underlying mechanism authors could re-analyze their PHF3 IP-MS data. Are known regulators of the RNA decay pathway among the interactors? The authors can address this with the data that they already have.

PHF3 interacts with many different RNA processing factors, which could mediate RNA stability regulation. We have now included a paragraph in the Discussion where we mention different interactors and how they could potentially contribute to the observed phenotype.

‘PHF3 interacts with alternative splicing factors such as SON, ZNF326, SAFB, RBMX and RBM15 (Fig. 1a), but the analysis of RNA-seq data did not reveal any major changes in splicing due to PHF3 loss. PHF3 interaction with RBM15, which is part of the m6A writer complex, may affect m6A RNA levels and thereby modulate mRNA stability⁸². Furthermore, PHF3 interacts strongly with SPT6, which was shown to regulate mRNA stability through the CCR4-NOT complex in yeast⁸³. CCR4-NOT promotes transcription elongation and deadenylates mRNAs as the first step in mRNA decay. Given that SPT6 facilitates CCR4-NOT recruitment⁸³, PHF3 might promote CCR4-NOT recruitment onto Pol II via SPT6 and thereby promote mRNA degradation.’

(3) The authors state ‘TFIIS ChIP-seq experiments showed increased TFIIS occupancy in PHF3 KO cells’. However, only TFIIS ChIP-seq data for two genes (GPR50 and GAPDH) are shown (Fig. 5f). Can the increase of TFIIS occupancy also be observed at other genes? What fraction of genes? A more global computational analysis would be helpful. Results of this computational analysis can be added to Fig. 5. Since no spike-in controls were used in the ChIP-seq experiments I suggest to soften this claim.

We have included a computational analysis of TFIIS ChIP-seq experiments for all genes classified according to upregulation in i) both PRO-seq and RNA-seq or ii) RNA-seq, which shows that TFIIS occupancy is increased on all genes that showed upregulation in both PRO-seq and RNA-seq. This supports the backtracking regulation model, whereby PHF3 negatively regulates transcription of a small subset of genes (PRO-seq and RNA-seq UP) through competition with TFIIS. This analysis is added to new Fig. 7f. We also indicated in figure legends that Pol II (F-12) and TFIIS ChIP-seq experiments were performed and normalized with mouse chromatin spike-ins.

Minor comments:

Introduction:

(1) It would be helpful for the reader to provide more information on known cellular functions of PHF3 in the introduction section. At the moment the authors only mention the family to which PHF3 belongs and then provide a more detailed description of the domain architecture.

This manuscript provides the first characterization of the cellular functions of PHF3, which is why we could not cite any relevant literature in the introduction section.

(2) Second paragraph, page 2: The authors provide a very nice overview of CTD interactors. It would be helpful if the authors could add clarification on whether these interactions refer to CTD Ser2-P only and indicate for which organism these interactions were detected. I guess that most of these studies were performed in yeast.

This information is now included in the Introduction: ‘Yeast and mammalian 5’ mRNA capping enzymes such as Cgt1, Pce1 and Mce1 were shown to employ the nucleotidyltransferase (NT) domain to directly bind pS5 within the Pol II CTD, whereas 3’end processing and termination factors such as yeast Pcf11 and mammalian SCAF8 employ the CTD-interaction domain (CID) to directly bind the pS2 mark on the Pol II CTD²⁴⁻²⁸.’

Similarly, on pages 13 (Discussion section), second paragraph, please clarify if the listed CTD reader domains that are highlighted refer to Ser2-P binding only and to indicate the organisms.

This information is now included in the Discussion:

‘Specific recognition of phosphorylated Serine-2 of the Pol II CTD classifies PHF3 SPOC as a new CTD reader domain, joining the company of (i) the CTD-interaction domain (CID) of yeast Rtt103 (pS2pS7, pT4), Pcf11 (pS2), Nrd1 (pS5), mammalian SCAF8 (pS2) and RPRD1A/B (pS2, pS7), (ii) the FCPH domain of mammalian Scp1 (pS5), (iii) the nucleotidyltransferase (NT) domain of yeast Cgt1, Pce1 and mammalian Mce1 (pS5), and (iv) the WW domain of mammalian PIN1 (pS2pS5)²².’

Results section:

(3) It is not clear to me why the authors performed the PHF3 IP-MS experiments for the overexpression FLAG-PHF3 line and have not used their cell line with endogenous GFP-tagged PHF3. A clarification would be helpful.

During the initial stage of the project when we performed mass spectrometry analysis, we did not have a GFP-tagged PHF3 cell line. To validate the interactions initially identified by FLAG-PHF3 overexpression, we later used the PHF3-GFP cell line with endogenous expression of PHF3.

(4) Page 4: ‘PHF3 SPOC did not bind the unphosphorylated CTD (...) but phosphorylation of S2 within both repeats was sufficient to confer strong binding.’ How strong was the binding μM or nM range? Adding values, if available, would be helpful.

Thank you for this remark. Although the dissociation constants (Kds) were included in the plots in Figure 2 and S2, we agree that they should also be included in the manuscript text.

‘Comparable affinity of PHF3 SPOC towards 2xpS2 ($K_d=1.6\pm 0.3 \mu\text{M}$), 2xpS2pS7 ($K_d=0.8\pm 0.1 \mu\text{M}$) and 2xpS2pS5 ($K_d=4.8\pm 0.3 \mu\text{M}$), coupled with lower affinity for 2xpS5 ($K_d=20.0\pm 4.0 \mu\text{M}$) or 2xpS7

(Kd=26.0±2.9 μM), suggested that PHF3 SPOC preferentially binds 2xpS2 (Fig. 2b,c and Extended Data Fig. 2a-f).'

(5) Page 10: 'We observed a significant overall increase in mRNA half-lives (...)'. What is the median increase in min or hours upon PHF3 or SPOC deletions? Please add this information to the main text.

Median half-lives are 5.948694 h for PHF3 KO, 5.970330 h for ΔSPOC and 4.837077 h for WT. This information is now included in the manuscript text.

(6) How is the correlation between SLAM-seq and PRO-seq replicate measurements? Adding correlation coefficients of replicate measurements to the main text or a statement to indicate the degree of correlation would clarify this.

The SLAM-seq samples were prepared in six biological replicates for each time point in each condition (3 conditions x 3 timepoints x 6 biological replicates). The high number of replicates enabled us to detect and remove low quality samples from the analysis. After filtering, the median correlation coefficient of conversion rates between samples belonging to the same group (samples belonging to the same condition and timepoint) was 0.75. The correlation heatmap is now included in Extended Data Fig. 11c.

The PRO-seq samples were prepared in three biological replicates for three different genotypes (PHF3 WT, KO, ΔSPOC). Mean Pearson correlation coefficient between the samples was 0.96, ranging from 0.95-0.97 for different gene parts. The correlation heatmap is now included in Extended Data Fig. 9c.

Discussion section:

(7) Page 13: 'While the Pol II CTD is the primary anchoring point for PHF3 (...)'. I agree that based on the data presented the CTD plays an important role in PHF3 localization to active transcription units, but CTD-independent interactions with Pol II or with elongation factors might be equally important in cells. Therefore, I suggest to soften this claim.

We replaced 'primary anchoring point' with 'docking point'. However, the fact that ΔSPOC phenocopies PHF3 KO and that PHF3 interactions with elongation factors such as SPT5, SPT6 and PAF1C are dependent on SPOC-phCTD interaction suggests that docking onto phCTD is indeed critical for PHF3 function in transcription regulation.

(8) Minor comments regarding Figure panels:

Fig. 5a and b is quite complex.

We agree but unfortunately we could not come up with a better way to present this data. This data has been moved to new Extended Data Fig. 14.

Fig. 6a: can both curves been presented in the same plot?

We have now combined the two curves in one plot.

Fig. 6e, f: please add labels for wt, dPHF3, dSPOC

Thank you, we added the labels.

Fig. 7f: GO terms are hard to read. Increasing font sizes would be helpful.

Thank you, we increased the font size.

REVIEWERS' COMMENTS

Reviewer #2 (Remarks to the Author):

The authors have adequately addressed my previous comments.

Reviewer #3 (Remarks to the Author):

The authors have addressed all of my comments. Most notably, the authors performed additional experiments and computational analyses to convincingly show that PHF3 acts as a general and positive regulator of Pol II pause release. The new data is presented in Figure 5. Moreover, the authors now provide more information on the potential molecular mechanisms of how PHF3 can modulate RNA stability. The additional experiments and the clarifications in the main text have further improved an already strong manuscript. Overall, the authors have done an exceptional job.